# *Escherichia coli* FtsA forms lipid-bound minirings that antagonize lateral interactions between FtsZ protofilaments

Marcin Krupka[1,*], Veronica W. Rowlett[1,*,†], Dustin Morado[2], Heidi Vitrac[3], Kara Schoenemann[1], Jun Liu[2] & William Margolin[1]

Most bacteria divide using a protein machine called the divisome that spans the cytoplasmic membrane. Key divisome proteins on the membrane's cytoplasmic side include tubulin-like FtsZ, which forms GTP-dependent protofilaments, and actin-like FtsA, which tethers FtsZ to the membrane. Here we present genetic evidence that in *Escherichia coli*, FtsA antagonizes FtsZ protofilament bundling *in vivo*. We then show that purified FtsA does not form straight polymers on lipid monolayers as expected, but instead assembles into dodecameric minirings, often in hexameric arrays. When coassembled with FtsZ on lipid monolayers, these FtsA minirings appear to guide FtsZ to form long, often parallel, but unbundled protofilaments, whereas a mutant of FtsZ (FtsZ*) with stronger lateral interactions remains bundled. In contrast, a hypermorphic mutant of FtsA (FtsA*) forms mainly arcs instead of minirings and enhances lateral interactions between FtsZ protofilaments. Based on these results, we propose that FtsA antagonizes lateral interactions between FtsZ protofilaments, and that the oligomeric state of FtsA may influence FtsZ higher-order structure and divisome function.

[1] Department of Microbiology and Molecular Genetics, McGovern Medical School, 6431 Fannin Street, Houston, Texas 77030, USA. [2] Department of Pathology and Laboratory Medicine, McGovern Medical School, 6431 Fannin Street, Houston, Texas 77030, USA. [3] Department of Biochemistry and Molecular Biology, McGovern Medical School, 6431 Fannin Street, Houston, Texas 77030, USA. * These authors contributed equally to this work. † Present address: Centers for Disease Control and Prevention, 1600 Clifton Road NE, Mailstop G32, Atlanta, Georgia 30329, USA. Correspondence and requests for materials should be addressed to W.M. (email: William.Margolin@uth.tmc.edu).

Escherichia coli divides by binary fission using a machine of membrane-associated proteins at the midcell division site. This machine, referred to as the divisome, is required to synthesize a ring of septal peptidoglycan that ultimately splits the cell into two[1]. During the initial stages of divisome formation, FtsZ, a homologue of tubulin, and FtsA, a homologue of actin, help to assemble the initial proto-ring at the inner surface of the cytoplasmic membrane[2]. FtsA and another protein called ZipA act together to anchor FtsZ to the cytoplasmic membrane in E. coli[3,4]. Nonessential accessory proteins including ZapA, ZapC and ZapD bind to FtsZ and stabilize the proto-ring[5]. The proto-ring subsequently recruits the remainder of the divisome to the membrane, followed by septum synthesis and cytokinesis.

Recent studies have shed some light on how FtsZ and FtsA proteins assemble into a functional proto-ring. Super-resolution microscopy of several species of bacteria has revealed that proto-ring proteins localize in a patchy pattern at midcell[6–8]. It is well established that FtsZ assembles into dynamic GTP-dependent protofilaments, and single-molecule or polarization microscopy experiments suggest that these protofilaments are loosely arranged around the cell membrane in an ∼100 nm-wide zone at midcell[9–12]. These patches are dynamic, as they have been shown to move in Bacillus subtilis[7], and the FtsZ and FtsA structures within them are dynamic as well, with rapid subunit turnover as measured by photobleaching experiments[13,14]. Recently, FtsZ polymers have been shown to treadmill both in vitro and in vivo, explaining their dynamic turnover[15–17].

Although FtsA has important roles in tethering FtsZ to the membrane and recruiting later-stage divisome proteins[18–20], it is likely that FtsA also has a regulatory role in the proto-ring and later stages. Some clues have come from gain-of-function mutants of FtsA such as FtsA* that can accelerate cell division and bypass the need for some divisome proteins, including ZipA[21,22]. The current model, based on several lines of evidence including the location of FtsA*-like mutants near the FtsA dimer interface, is that while FtsA forms oligomers along the membrane that interact with FtsZ, FtsA* is more monomeric. According to the model, FtsA initially assembles as oligomers that are unable to recruit downstream divisome proteins. Divisome proteins such as ZipA, FtsX and/or FtsN then promote a conversion of FtsA to its FtsA*-like monomeric state, freeing the 1C subdomain of FtsA that is normally involved in oligomeric interactions to recruit late divisome proteins, leading to septum synthesis[22–25]. Recently, other potential divisome checkpoints have been uncovered by gain-of-function mutants, including mutations in later divisome genes such as ftsL[26,27] and even a mutation in ftsZ, L169R (henceforth called FtsZ*) that can also bypass ZipA[28].

FtsZ* is relevant for our understanding of the proto-ring for several reasons. First, in contrast to the single protofilaments formed by FtsZ, purified FtsZ* mostly assembles into paired protofilaments, suggesting that it bypasses ZipA by mimicking ZipA's ability to bundle FtsZ protofilaments[28,29]. For simplicity, we will henceforth use the term 'bundling' to denote increased lateral interactions between adjacent FtsZ protofilaments. The increased bundling of FtsZ* is likely a result of increased lateral interactions between subunits, as its L169R mutation is at the side of the FtsZ subunit. Second, although excess cellular levels of FtsA are well known to inhibit cell division, FtsZ* resists this inhibition much more robustly than wild-type (WT) FtsZ[28], prompting us to hypothesize that WT FtsA inhibits FtsZ protofilament bundling. Consistent with this, a recent study of the in vitro assembly of FtsA and FtsZ on supported lipid bilayers showed that treadmilling FtsZ polymers were more dynamic when tethered to the bilayers by FtsA than by ZipA, suggesting that FtsA promotes turnover of FtsZ subunits[17].

Here, we provide genetic and biochemical evidence supporting that FtsA inhibits FtsZ protofilament bundling in E. coli. Using conditions or mutants that promote or inhibit FtsZ bundling, our results indicate that FtsA antagonizes FtsZ protofilament bundling in vivo. We then show that, unlike FtsA proteins from species such as Streptococcus and Thermotoga that form long polymers in vitro[30–33], E. coli FtsA assembles into arrays of minirings on lipid monolayers. In contrast, FtsA* rarely forms minirings and instead forms smaller oligomers. We then show that, in vitro, a carpet of FtsA minirings promotes FtsZ assembly into long, often parallel protofilaments, whereas the more randomly oriented and densely packed FtsA* oligomers on the membrane promote assembly of bundled FtsZ protofilaments, apparently free of the constraints imposed by the minirings. In addition to providing biochemical evidence that FtsA* is less oligomeric than FtsA, these results support that FtsA may be sufficient to antagonize the lateral interactions between FtsZ protofilaments in vitro.

## Results

**FtsA antagonizes excess FtsZ protofilament bundling in vivo.** It is well known that an excess of FtsA inhibits E. coli cell division that can be corrected by a compensatory increase in FtsZ[34,35]. Our previous observation that the hyperbundled FtsZ* mutant, even at native levels, can resist the toxicity of excess FtsA[28] prompted us to hypothesize that FtsA is normally an antagonist of FtsZ protofilament bundling; because FtsZ* protofilaments bundle more strongly, they can resist the anti-bundling activity of FtsA. If so, then increasing FtsA levels should rescue FtsZ overbundling by extrinsic factors, and decreasing FtsZ bundling should exacerbate FtsA toxicity.

We decided first to explore the effects of FtsA on the toxicity triggered by excess ZapA protein that is known to organize FtsZ protofilaments in bundles and crosslink them non-covalently[36–38]. Notably, ZapA overproduction leads to cell elongation with abnormal bends and mislocalized FtsZ rings[39], presumably because of excess FtsZ protofilament bundling. If our model is correct, FtsA should counteract these deleterious effects. To examine the phenotype of WT E. coli cells co-overproducing ZapA and FtsA, we induced zapA expression from a sodium salicylate-inducible plasmid, pKG116-zapA, and simultaneously induced expression of ftsA from the compatible plasmid pDSW210F containing an isopropyl-β-D-thiogalactoside (IPTG)-inducible P_{trc} promoter.

As expected, excess ZapA from induced pKG116-zapA reduced viability at least 10-fold on serial dilution plates of exponentially growing cells carrying the pDSW210 empty plasmid, either with 0 or 50 μM IPTG (Fig. 1a). Consistent with the previous ZapA overproduction phenotype, many of these cells were filamentous and some displayed abnormal twists with mislocalized FtsZ that formed random accumulations instead of midcell rings (Fig. 1a). In contrast, in cells with pDSW210F-ftsA, even leaky expression from the uninduced P_{trc} promoter was sufficient to restore cell viability on serial dilution plates, as was induction with 50 μM IPTG (Fig. 1a). Consistent with this improvement in viability by extra FtsA, cell filamentation and bending were largely suppressed, and positioning of FtsZ rings at midcell was largely restored. Levels of overproduced ZapA were the same whether or not FtsA was induced (Supplementary Fig. 1).

As an independent way to show that FtsA antagonizes FtsZ bundling in vivo, we tested the E93R mutant of FtsZ that was previously shown to increase lateral interactions between protofilaments and had a deleterious effect on FtsZ function in vivo[40]. As with the ZapA co-overproduction experiments described above, FtsZ_{E93R} was produced from a salicylate-

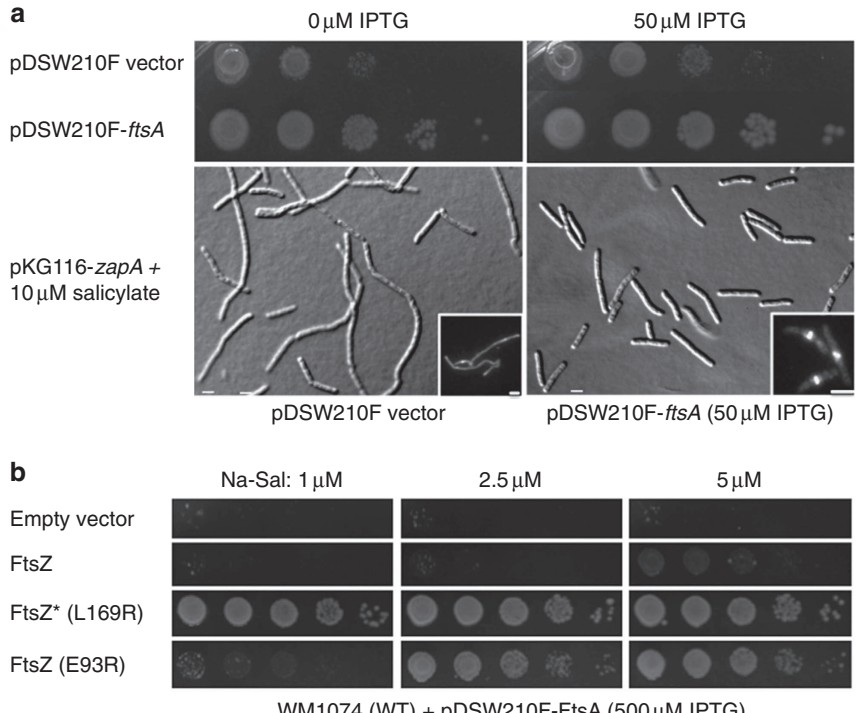

**Figure 1 | Suppression of ZapA overproduction toxicity by FtsA.** (**a**) WM1074 cells harbouring pKG116-ZapA and either pDSW210F or pDSW210F-FtsA were spot diluted on plates containing 10 μM sodium salicylate and 0 or 50 μM IPTG to induce pKG116 and pDSW210F, respectively. Corresponding differential interference contrast (DIC) images of WM1074 cells co-overproducing ZapA and pDSW210F constructs are shown below. Insets show immunofluorescence microscopy (IFM) of FtsZ. Scale bars, 2 μm. (**b**) WM1074 cells harbouring pDSW210F-FtsA and either pKG110, pKG110-FtsZ, pKG110-FtsZ* or pKG110-FtsZ_{E93R} were spot diluted on plates containing the indicated levels of sodium salicylate and 500 μM IPTG to induce pKG110 derivatives and pDSW210F-FtsA, respectively.

inducible plasmid, except we used pKG110 that has a weaker ribosome-binding site than pKG116 for finer control over induction levels. Normally, FtsA from pDSW210F-*ftsA* produced at the high level of 500 μM IPTG is toxic to cells (Fig. 1b, row 1). Co-producing WT FtsZ from pKG110-*ftsZ* at 1 or 2.5 μM salicylate could not restore viability to cells with this level of FtsA, and even 5 μM salicylate could only restore weak growth (Fig. 1b, row 2). In contrast, FtsZ_{E93R} was able to restore cell viability, although this effect was weaker than that for FtsZ*, as 2.5 μM or higher salicylate induction was required for viability instead of 1 μM (Fig. 1b, rows 3 and 4). FtsZ_{E93R} protein was produced at levels similar to FtsZ and FtsZ* after induction with salicylate (Supplementary Fig. 2), confirming that its resistance to excess FtsA was a result of intrinsic activity of the protein and not higher cellular levels than WT FtsZ.

**Deletion of *zapA* and/or *zapC* exacerbates FtsA toxicity.** We then asked whether, conversely, FtsA was more toxic in cells lacking Zap proteins. Deletions of single *zapA*, *zapC* or *zapD* genes do not affect cell viability, but result in moderate cell filamentation; deletion of multiple *zap* genes inhibits cell division more severely, probably because of insufficient bundling of FtsZ[41]. Therefore, if FtsA inhibits FtsZ polymer bundling, excess FtsA should be more toxic in a Δ*zapA* strain compared with the *zapA* + parent strain, and even more toxic in a Δ*zapA* Δ*zapC* strain[41].

Modest FtsA overproduction from pDSW210F-*ftsA* at 50 μM IPTG was not toxic in the WT TB28 parent strain, with normal viability on serial dilution plates (Fig. 2a), normal cell length (Supplementary Fig. 3) and FtsZ rings at midcell (Fig. 2b). In contrast, the same induction level of pDSW210F-*ftsA*, which resulted in similar cellular FtsA levels as TB28 (Supplementary

Fig. 4), severely affected cell growth and viability of the Δ*zapA* strain (Fig. 2a,c). Many of these cells elongated into filaments that were much longer than the few filaments of the Δ*zapA* parent or Δ*zapA* strain with uninduced pDSW210F-*ftsA* (Supplementary Fig. 3). Nevertheless, many of the long filaments retained multiple FtsZ rings or zones (Fig. 2c).

As expected, the Δ*zapA* Δ*zapC* double mutant on its own exhibited a range of filamentous cells, although most of the cells were under 10 μm in length (Supplementary Fig. 3). Induction of pDSW210F-*ftsA* in these cells with either 0 or 50 μM IPTG yielded cellular levels of FtsA similar to those in TB28 or Δ*zapA* cells at the same induction levels (Supplementary Fig. 4). However, the cell division inhibition by FtsA was significantly exacerbated in the double mutant: induction with 50 μM IPTG prevented all growth on plates and the already filamentous phenotype of Δ*zapA* Δ*zapC* cells became much more extreme (Fig. 2a,d and Supplementary Fig. 3). These very long filaments were largely devoid of FtsZ rings, although periodic diffuse areas of fluorescence that might reflect attempted FtsZ ring assembly were still visible in some cells (Fig. 2d). Importantly, FtsZ levels in cells overproducing FtsA were not changed with respect to uninduced cells after normalizing for cell density, indicating that excess FtsA does not decrease cellular levels of FtsZ (Supplementary Fig. 4). Together, the ability of excess FtsA to correct the toxicity of excess ZapA, yet worsen cell division problems in the absence of ZapA and/or ZapC, suggests that FtsA has the capacity to antagonize the FtsZ bundling activities of ZapA and ZapC *in vivo*.

**FtsA requires membrane attachment to inhibit FtsZ bundling.** Considering that WT *E. coli* FtsA was active on supported lipid

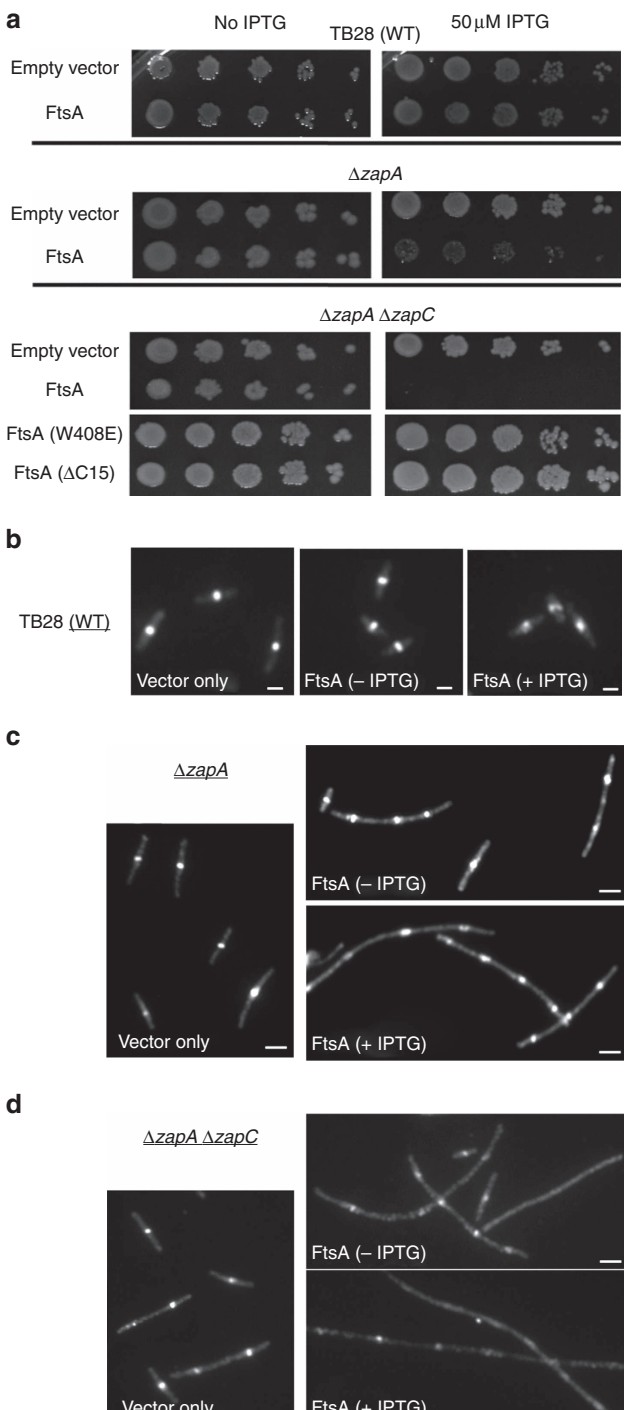

**Figure 2 | Excess FtsA exacerbates the loss of *zapA/zapC*. (a)** Spot dilutions of TB28, *ΔzapA* and *ΔzapA ΔzapC* cells transformed with pDSW210F, pDSW210F-FtsA, pDSW210F-FtsA(W408E) or pDSW210F-FtsA(ΔC15) and induced with 0 or 50 μM IPTG. **(b)** Immunofluorescence microscopy (IFM) of FtsZ in TB28 cells transformed with pDSW210F vector, uninduced or induced pDSW210F-FtsA. **(c)** IFM of FtsZ in *ΔzapA* and **(d)** *ΔzapA ΔzapC* cells transformed with indicated uninduced ( − IPTG) or induced ( + IPTG) pDSW210F constructs. Scale bars, 2 μm.

bilayers[17] and had no effect on FtsZ in solution[42], we reasoned that FtsA must be membrane bound to have any relevant effect on FtsZ. To test this idea, we used two mutants of FtsA that have a defective membrane-binding helix: W408E, which disturbs the amphipathic nature of the helix, and ΔC15, a complete truncation

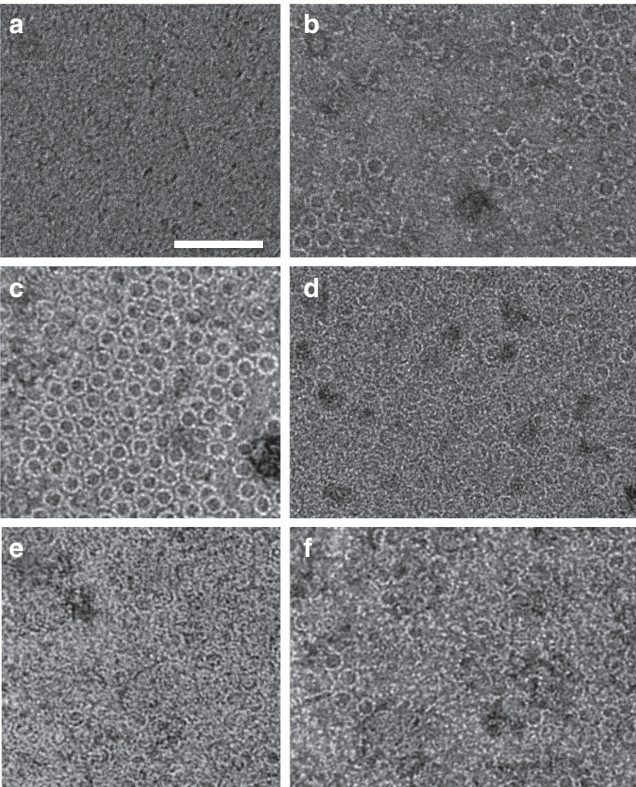

**Figure 3 | FtsA minirings on lipid monolayers visualized by negative stain transmission electron microscopy. (a)** Monolayer with buffer + ATP but no FtsA; **(b)** 0.1 μM FtsA + ATP, showing miniring arrays; **(c)** 0.5 μM FtsA + ATP, showing a larger FtsA miniring array; **(d)** 1 μM FtsA + ATP; **(e,f)** 0.5 μM FtsA with no added ATP **(e)** or with ATP **(f)** from the same experiment. Scale bar, 100 nm.

of this domain[18,43]. Notably, when these FtsA mutant proteins were overproduced in a *ΔzapA ΔzapC* strain under the same induction conditions (50 μM IPTG) that prevented colony formation with WT FtsA, we observed normal colony viability (Fig. 2a, bottom rows). These results suggest that FtsA must have an intact membrane-binding helix and consequently be membrane bound to exert its antagonistic effects on FtsZ polymer bundling. This is consistent with previous conclusions about the need for FtsA membrane attachment in FtsA–FtsZ interactions[18].

**FtsA assembles into minirings on lipid monolayers.** The requirement for membrane attachment to antagonize FtsZ bundling *in vivo* prompted us to explore this inhibitory effect *in vitro* on lipid monolayers. To ensure that both FtsA and FtsZ could function in this purified system, we used physiological concentrations of FtsA (500–1,000 molecules per cell, or 0.5 μM) and FtsZ (3,000-7,000 molecules per cell, or 2.5–5 μM)[44].

As expected, no structures were visible on the monolayers by negative stain transmission electron microscopy without added protein (Fig. 3a). Strikingly, however, we found that FtsA assembled into minirings of ∼ 20 nm diameter, the abundance of which was roughly proportional to FtsA concentration (Fig. 3b–d). These FtsA minirings were similar in size to minirings formed as curved extensions of straight FtsZ protofilaments assembled on cationic lipid monolayers[45]. Nevertheless, the FtsA minirings were distinct in several ways: they seem not to be associated with straight filaments, are more uniform in size and appear to have a strong tendency to

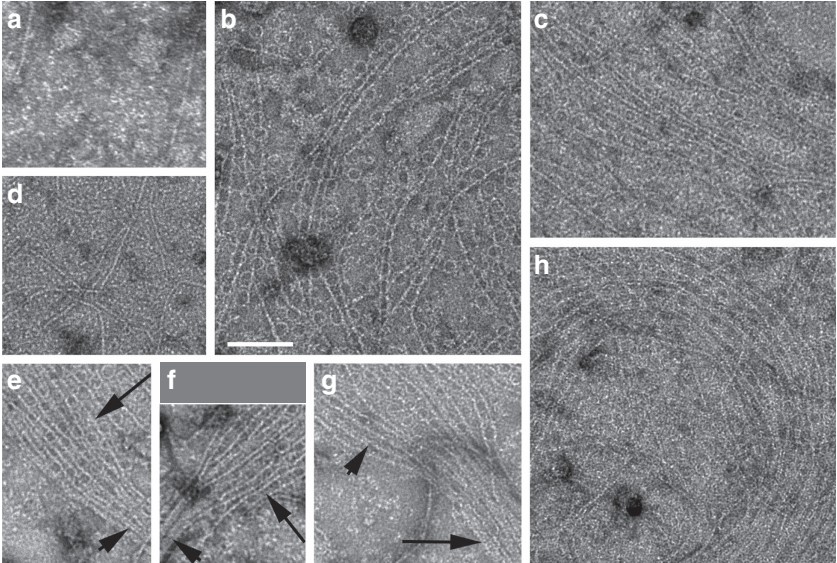

**Figure 4 | Assembly of FtsZ on monolayers seeded with FtsA.** (**a**) Mock experiment in which 5 μM FtsZ is added to monolayers, but no FtsA was present; (**b**) 0.1 μM FtsA + 5 μM FtsZ; (**c**) 0.5 μM FtsA + 5 μM FtsZ; (**d**) 0.5 μM FtsA + 5 μM FtsZ on a grid without a monolayer. (**e–g**) Same conditions as (**c**), showing examples of aligned unbundled FtsZ filaments (long arrows) on FtsA minirings merging with bundled FtsZ filaments not associated with visible FtsA minirings (short arrows). (**h**) A large swirl of aligned FtsZ filaments on FtsA minirings. ATP and GTP were added to all experiments. Scale bar, 100 nm.

self-interact. Even at the lowest FtsA concentration of 0.1 μM, the minirings tended to form small arrays, suggesting that lateral interactions between minirings occur in addition to longitudinal interactions between subunits (Fig. 3b). In addition to the minirings, we noticed many incomplete minirings or arcs, in which the FtsA oligomers maintained similar curvature, although occasional arcs had distorted curvature (Fig. 3b). The 1 μM concentration of FtsA often produced areas of too many overlapping minirings and arcs (Fig. 3d), and hence we generally used 0.5 μM (Fig. 3c) in subsequent experiments to distinguish them more easily. Although ATP was added to the mixtures, we found that FtsA minirings assembled without adding exogenous ATP, and addition of more ATP had no reproducible effect on miniring assembly (Fig. 3e,f). This is consistent with the presence of ATP in the FtsA protein preparation that was proposed previously[32]. As might be expected, the degree of miniring assembly varied between experiments and from one area of the monolayer to another, and hence we derived our conclusions from trends observed in many images and experiments, and made our best effort to show typical images.

**FtsA minirings align FtsZ polymers and block their bundling.** At physiological protein concentrations, soluble FtsZ protein forms randomly oriented, single and sometimes bundled protofilaments as seen by transmission electron microscopy[45,46]. We observed the same when FtsZ was incubated with electron microscopy grids covered by a lipid monolayer, using the same buffer also used for FtsA–FtsZ experiments. Although FtsZ does not interact significantly with lipids, there was sufficient FtsZ in the solution that some protofilaments ended up scattered on the grid (Fig. 4a). When FtsZ and FtsA were added to a regular grid without lipids, FtsZ filaments that stuck to the grid were readily observed, but no minirings were present (Fig. 4d). In contrast, when FtsZ was added to lipid-bound FtsA minirings, FtsZ assembled into protofilaments that were hundreds of μm long and strikingly aligned (Fig. 4b,c). When FtsA minirings covered large swaths of the monolayer, it was difficult to discern how FtsZ polymers associated with them. However, in areas

where minirings were more limited, the vast majority of aligned FtsZ protofilaments associated with FtsA minirings that were in turn roughly aligned with the FtsZ protofilaments.

The most remarkable characteristic of these aligned FtsZ protofilaments was that they did not form bundles. These unbundled protofilaments varied in spacing, but were usually significantly > 10 nm apart as judged by negative stain intensity profiles (Fig. 4b,c and Supplementary Fig. 5a,b). These distances are distinct from the spacing between FtsZ protofilaments under bundling conditions (see below) of ~ 5–8 nm (Supplementary Fig. 5c–d). We also observed many examples of aligned but unbundled FtsZ protofilaments on a carpet of FtsA minirings that merged into highly bundled protofilaments in an area with no visible FtsA minirings (Fig. 4e–g). Large swirls of aligned FtsZ protofilaments (Fig. 4h) were reminiscent of the micron-sized swirls observed previously[17]. Taken together, these results suggest that membrane-bound FtsA minirings align FtsZ protofilaments and prevent their bundling.

To determine how quickly the above organization occurred, we added FtsZ to lipid-bound FtsA minirings as described above, triggered FtsZ assembly with addition of GTP and then took samples after short time intervals. After only 30 s following addition of GTP, most fields of view featured FtsA rings, on top of which were unbundled FtsZ protofilaments of variable length that were not all aligned (Supplementary Fig. 6a). This pattern was similar after 1 min. However, after 5 min of incubation and thereafter, the FtsZ protofilaments on FtsA minirings became mostly aligned and resembled the patterns shown in Fig. 4b,c,h, and Supplementary Fig. 6b. We interpret these results to suggest that treadmilling of FtsZ on the carpet of FtsA reorganized the FtsZ filaments and reinforced their parallel alignment. We also observed similar patterns of FtsA minirings overlaid with unbundled FtsZ protofilaments when both FtsA and FtsZ were added simultaneously to the lipid monolayers, indicating that FtsA does not require a separate preassembly step at the membrane to form minirings (Supplementary Fig. 7).

**The 3D tomography of FtsA minirings and FtsZ protofilaments.** For a more detailed exploration of the structures of the minirings

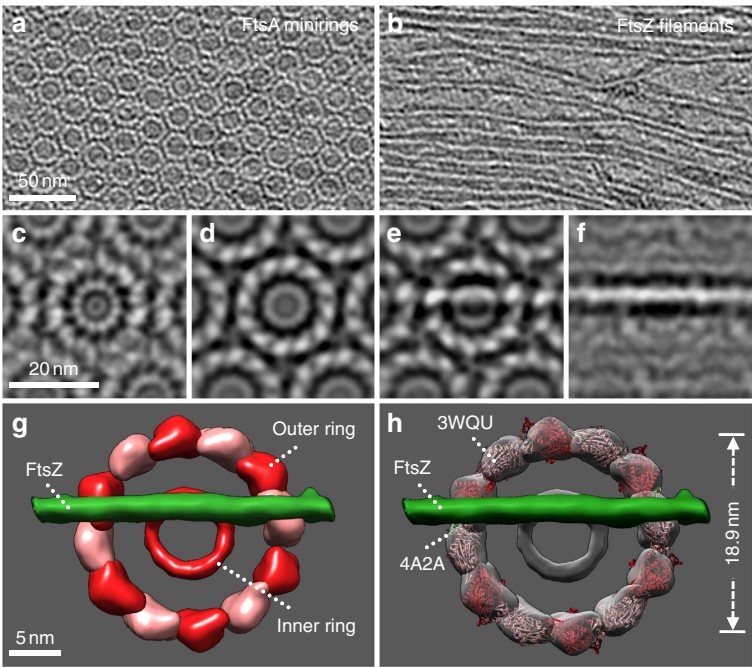

**Figure 5 | Tomography and structural modelling of FtsZ protofilaments aligned on top of FtsA minirings on lipid monolayers.** (**a,b**) Two tomographic slices of the same transmission electron microscopy (TEM field), from the lipid monolayer upward, showing a FtsA miniring hexagonal array (**a**), then aligned unbundled FtsZ protofilaments on top of the array (**b**) (see also Supplementary Movie 1). (**c–f**) A subvolume-averaged portion of a field from (**a,b**), with tomographic slices from the lipid monolayer (**c**) up to FtsZ filaments (**f**). (**g**) Segmentation of the 3D reconstructions showing potential sites of interaction where an FtsZ filament (green) intersects with the FtsA miniring (red and pink subunits). (**h**) Atomic structures from *S. aureus* FtsA (3WQU) are shown docked into the miniring densities, along with the FtsA-binding peptide from the C terminus of FtsZ (4A2A) (green) into the two subunits interacting with the crossing FtsZ protofilament. See also Supplementary Movie 2. For simplicity, the other 10 FtsZ C-terminal peptides are not shown.

and how they interact with FtsZ filaments on the membrane, we analysed negative stain images with arrays of uniform FtsA minirings and their associated aligned, unbundled FtsZ protofilaments by three-dimensional (3D) tomography (Supplementary Movie 1). Raw tomographic slices of FtsA–FtsZ assembly (Fig. 5a,b) as well as subvolume averaging of over 18,000 minirings (Fig. 5c–f) revealed that FtsA minirings consist of 12 subunits encircling a 10 nm-wide inner ring, also with 12-fold symmetry (Fig. 5c–e). This inner ring may represent a symmetrical assembly of FtsA membrane targeting helices corresponding to each of the 12 miniring subunits, because it was most visible in the membrane-proximal tomographic slice (Fig. 5c).

To determine how individual FtsA monomers might interact in such a structure, we docked the published FtsA atomic structure from *Staphylococcus aureus*[47,48], into the densities. The docking model fitted 12 subunits into the miniring structure as a hexamer of dimers (Fig. 5h and Supplementary Movie 2), with a 60° twist between each monomer, arranged head to tail as proposed previously[32,49,50]. The spacing between two adjacent subunits is 4.95 nm, similar to the ∼5 nm spacing between other monomers of actin and actin homologues[51].

Analysis of raw tomographic slices showed that FtsZ protofilaments were located above the FtsA minirings, consistent with the FtsZ membrane-tethering role for FtsA (Fig. 5b and Supplementary Movie 1). Each crossing FtsZ protofilament seems to interact with two FtsA subunits at its two points of intersection with the miniring (Fig. 5e–h). Notably, the interaction site on FtsA for the C-terminal peptide of FtsZ (PDB:4A2A)[32] faces upward, away from the membrane, and subunits that seem to interact directly with FtsZ are highlighted in Fig. 5h. Although higher-resolution structures will be needed to confirm the model, it is consistent with the binding of an FtsZ protofilament to

specific FtsA subunits in the miniring that could potentially explain how protofilaments are aligned and bundling is inhibited.

**FtsA\* forms short curved filaments that allow FtsZ bundling.** FtsA\* is believed to self-interact less than WT FtsA, and this property strongly correlates with its gain-of-function activities *in vivo*[22]. However, decreased FtsA\* oligomerization was inferred by *in vivo* assays that were largely based on deletions of the membrane targeting helix or fusions to other protein domains. To determine directly whether FtsA\* has an assembly defect, we tested the assembly of purified FtsA\* on lipid monolayers.

Notably, we found that FtsA\* seldom formed complete minirings and instead assembled numerous tightly packed arcs and short filaments, even at concentrations (0.5 μM) where WT FtsA formed mostly minirings (Fig. 6b). Importantly, at a concentration of 0.1 μM, where FtsA still formed arcs and minirings, FtsA\* failed to form minirings or detectable arcs, instead forming mostly short heterogeneous structures on the monolayer that were difficult to define (Fig. 6a). These data strongly suggest that FtsA\* can assemble into curved oligomers like FtsA, but does so less efficiently, thus supporting previous genetic data. Moreover, in contrast to WT FtsA, lipid-bound FtsA\* allowed FtsZ protofilaments to bundle and to form potential crosslinks. For example, most of the FtsZ polymers on lipid-bound FtsA\* appeared as bundles of protofilaments, spaced ∼7–8 nm apart (Fig. 6c,d and Supplementary Fig. 8). These data suggest that FtsA\* may not be able to antagonize lateral interactions between FtsZ protofilaments and instead may promote them.

The images of FtsA\* alone (Fig. 6b) or when together with FtsZ (Supplementary Fig. 9) suggest that FtsA\* oligomers are tightly packed on the monolayers. Nevertheless, to support the idea that the deficiency of FtsA\* minirings stems from reduced monomer–monomer interactions on the membrane and not

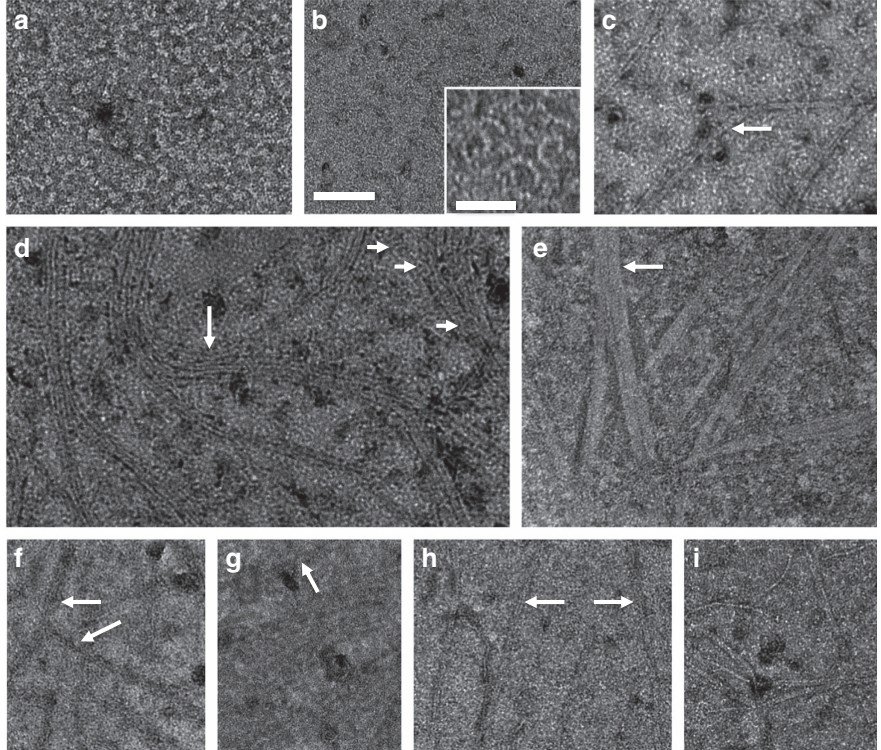

**Figure 6 | Assembly of FtsA\* and FtsZ\* on lipid monolayers and their effects on FtsZ and FtsA higher-order assembly.** (**a**) The 0.1 μM FtsA\*, showing short straight polymeric structures; (**b**) 0.5 μM FtsA\*, showing mostly arcs and short curved protofilaments. Inset shows × 2.5 magnification of a portion of the image, highlighting some FtsA\* arcs (scale bar in inset, 40 nm). (**c**) The 0.1 μM or (**d**) 0.5 μM FtsA\* + 5 μM FtsZ, showing bundling of FtsZ protofilaments (large arrow) and FtsA\* arcs (smaller arrows). (**e**) The 0.5 μM FtsA\* + 5 μM FtsZ\*, showing assembly of very large FtsZ protofilament bundles/sheets (arrow). (**f,g**) The 0.5 μM FtsA + 5 μM FtsZ\*, showing persistence of FtsZ\* bundles in the presence of FtsA minirings. (**h**) Same as (**f,g**) except with 2.5 μM FtsZ\*. Arrows highlight FtsZ\* protofilament bundles. (**i**) Mock experiment with FtsZ\* on monolayers, but no FtsA added, showing an unusually dense area with FtsZ\* polymers on the monolayer for comparison. Scale bar, 100 nm.

a lower concentration on the membrane because of a membrane-binding defect, we investigated the ability of FtsA, FtsA\* or FtsAΔC15 to cosediment with liposomes. Previous cell fractionation experiments indicated that FtsA\* was proficient at membrane binding, and could even suppress the membrane-binding defect of the W408E mutant in *cis*[43]. Consistent with these *in vivo* data, we found that FtsA and FtsA\* behaved essentially identically in their ability to cosediment with liposomes (Supplementary Fig. 10). These results, along with the high density of FtsA\* oligomers that we observed on lipid monolayers, confirm that FtsA\* is at least as proficient as FtsA in binding to lipid membranes.

**FtsZ\* remains bundled with FtsA and bundles more with FtsA\*.**
If FtsA minirings can antagonize FtsZ protofilament bundling, then we reasoned that the resistance of FtsZ\* to excess FtsA *in vivo*[28] could be caused by the ability of FtsZ\* to remain bundled even when tethered to the membrane by FtsA minirings. As with FtsZ, some FtsZ\* alone with no FtsA stuck to the monolayer as scattered protofilament doublets (Fig. 6i). At equivalent concentrations of FtsA that inhibited WT FtsZ bundling, FtsZ\* formed double filaments that were often aligned as variable width bundles of doublets (Fig. 6f–h, arrows). Therefore, unlike FtsZ, lateral interactions between FtsZ\* protofilaments persisted in the presence of WT FtsA. This is consistent with the ability of FtsZ\* to resist the toxic effects of FtsA *in vivo*. At very low concentrations (1 μM) of FtsZ\*, some paired FtsZ\* protofilaments were scattered on the lipid monolayer in the same general area as FtsA minirings, although the FtsA associated with the paired FtsZ\* protofilaments

was seldom in miniring form (Supplementary Fig. 11a). At higher, more physiological concentrations, FtsZ\* continued to form double protofilaments as expected, but FtsA minirings were replaced with short curved FtsA\*-like oligomers (Supplementary Fig. 11b). Together with the results in Fig. 6c,d and Fig. 4e–g, the apparent disruption of FtsA minirings with higher levels of FtsZ\* suggests that disruption of FtsA minirings enhances FtsZ protofilament bundling and that, in turn, increased FtsZ protofilament bundling (constitutive in the case of FtsZ\*) can disrupt FtsA minirings.

Individually, FtsZ\* or FtsA\* can bypass ZipA, and FtsA\* corrects many cell division defects[21,28]. However, when combined, FtsZ\* and FtsA\* exacerbate cell division, causing a cell twisting phenotype similar to that of ZapA overproduction[28]. Although the causes of this phenotype are likely complex, one possibility is that the intrinsic enhancement of lateral interactions by the FtsZ\* mutation is stimulated further by the interactions with FtsA\* and tips the balance too far. To test this notion in our purified system, we added the 0.5 μM FtsA\* to the monolayers and then added 5 μM FtsZ\*. Consistent with the *in vivo* phenotype, FtsZ\* formed very large bundles or sheets under these conditions that were many protofilaments wide (Fig. 6e and Supplementary Fig. 5c,d). This is another case of the purified system confirming inferences made from *in vivo* phenotypes, and sets the stage for future studies.

**Discussion**
Here we provide several lines of *in vivo* evidence that FtsA antagonizes the bundling of FtsZ protofilaments, suggesting that FtsA not only anchors FtsZ to the membrane, but also

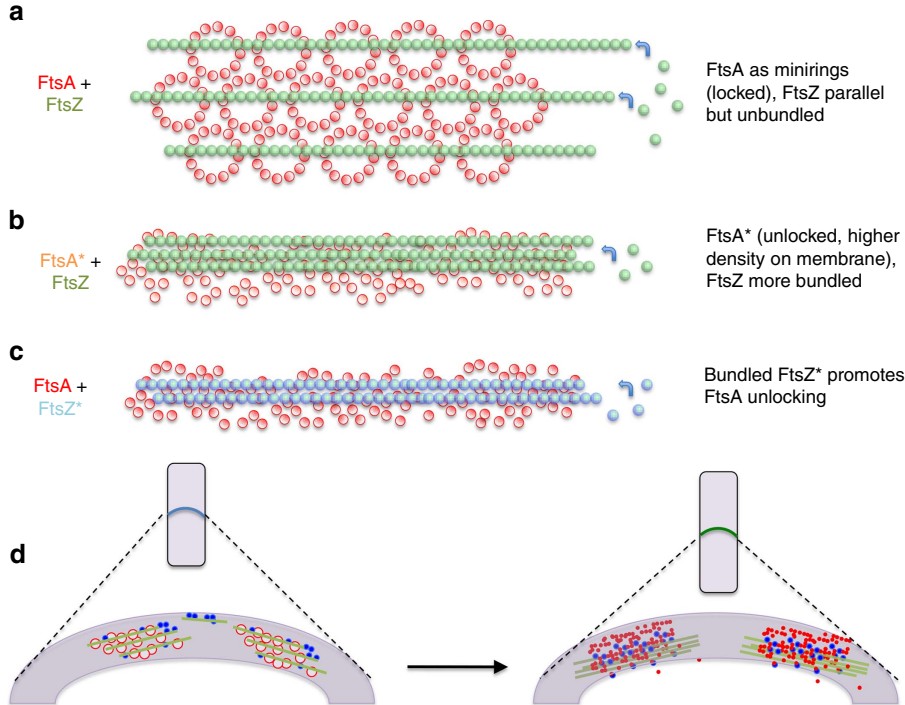

**Figure 7 | Proposed model for FtsA miniring functions.** We propose that FtsA minirings may align FtsZ protofilaments and inhibit their bundling until the minirings are disrupted. (**a**) FtsZ protofilaments (green) are initially kept apart in an unbundled state by membrane-bound FtsA minirings (red). This form of FtsA is toxic to cells when in excess because unbundled FtsZ cannot assemble into an active FtsZ division ring needed for later stages in septation. (**b**) FtsA* (orange) on lipid monolayers displays abundant curved oligomers but full minirings are rare, in agreement with genetic evidence indicating that FtsA* is in a more monomeric state than FtsA. Unlike FtsA, lipid-bound FtsA* promotes lateral interactions among FtsZ protofilaments (green), probably because FtsA* oligomers are more randomly and tightly packed on the membrane, removing the geometric constraints imposed by FtsA minirings on FtsZ protofilament assembly and promoting the natural tendency of membrane-tethered FtsZ protofilaments to associate laterally. We speculate that this 'unlocked' FtsA* is not toxic to cells when in excess because FtsZ bundling is needed for activity of the septal ring. Cells with FtsA* may bypass the initial step (**a**) and proceed directly to active septation. (**c**) Adding physiological levels of FtsZ* (blue) to lipid monolayers seeded with FtsA (red) results in FtsA*-like structures on the membrane and no minirings, suggesting that the protofilament pairs of FtsZ* promote FtsA miniring disassembly. This is consistent with the FtsA*-like phenotype of FtsZ* *in vivo*, including the ability to bypass ZipA. FtsZ treadmilling (blue arrows) may occur in all three scenarios. (**d**) The ability of FtsA* to promote bundling of FtsZ protofilaments and the ability bundled FtsZ* protofilaments to disrupt FtsA minirings suggests that the increased FtsZ bundling and decreased higher order structures of FtsA are self-reinforcing and therefore irreversible. Because FtsA* accelerates cell division *in vivo* and can suppress defects in other cell division proteins, we propose that bundling of FtsZ protofilaments (green) by conversion of FtsA minirings (red circles) into shorter FtsA oligomers (red), aided by ZipA and Zap proteins (blue), may be crucial for septal ring activity, and is consistent with evidence that FtsZ filaments become more dense as the Z ring constricts.

counterbalances the FtsZ bundling effects of the Zap proteins. These two activities of FtsA are coupled, as mutants of FtsA that are defective in membrane binding are also unable to antagonize FtsZ bundling *in vivo*. To obtain biochemical support for this activity, we asked whether purified FtsA could inhibit bundling of FtsZ on lipid monolayers. Remarkably, we found that purified FtsA forms ∼20 nm-wide minirings on the monolayers, and these minirings are associated with aligned but unbundled FtsZ protofilaments (Fig. 7a). These minirings are unexpected because FtsA proteins from several other bacteria have been shown to form straight or helical actin-like filaments in solution or on membranes[30–32]. Although *E. coli* FtsA has not yet been crystallized, docking of the *S. aureus* FtsA molecule into the miniring density suggests that subunits of *E. coli* FtsA interact with a twist, resulting in a head-to-tail oligomer that can close on itself to create a 12-membered miniring, essentially a hexamer of dimers. Further work would be needed to demonstrate the formation of FtsA minirings in *E. coli* cells.

When physiological concentrations of FtsA (0.5 μM) are added to lipid monolayers, the minirings interact to form large hexagonal arrays, and smaller arrays are present at 0.1 μM. Although there is no direct evidence that these miniring arrays are present in cells, the patches of FtsZ/FtsA localization in *E. coli* and other species observed by super-resolution microscopy may represent small arrays of FtsA rings overlaid with FtsZ protofilaments, perhaps similar to the organization shown in Fig. 5b,c. For example, FtsA is estimated at ∼900 molecules per 1 μm-thick or 3.1 μm-circumference *E. coli* cell in rich medium[44,52]. Assuming that most cellular FtsA is in the proto-ring, 6 arrays of 12 minirings each (864 FtsA monomers in total) could fit in a proto-ring with a circumference of ∼3.1 μm if the miniring clusters of ∼150 nm diameter each were spaced ∼400 nm apart. This is similar to what is observed in *E. coli* by super-resolution fluorescence microscopy[8].

In contrast to FtsA, FtsA* often assembles into incomplete minirings or arcs as well as other nonring structures (Fig. 7b). This is the first biochemical evidence that FtsA* is less able to oligomerize in a head-to-tail manner to form minirings, providing a molecular basis for previous genetic studies of FtsA* and other FtsA*-like mutants. Notably, 0.1 μM FtsA* generally fails to form arcs and instead forms short straight polymers that seem to be doublets. These structures may represent aberrant lateral interactions between oligomers that are unmasked when the normal head-to-tail oligomerization is weak. The nature and

significance of these structures is not yet clear. Nonetheless, the ability of FtsA* to bind to the membrane at least as well as FtsA and the tendency of FtsA* to form a mass of short oligomers on the membrane instead of minirings suggests that the two-dimensional packing density of FtsA* may be higher than that of FtsA. We therefore propose that the strong bundling of FtsZ protofilaments in the presence of FtsA* is caused by the higher packing density of membrane anchors (Fig. 7b).

We also propose that that FtsA oligomeric state not only can induce changes in FtsZ lateral interactions, but also may be affected by them. For example, we observed that the lack of FtsA minirings was strongly correlated with bundled WT FtsZ. Moreover, when FtsZ* was added to monolayers containing preassembled WT FtsA, FtsZ* continued to form mostly paired protofilaments on monolayers as expected, but most FtsA minirings were disrupted, becoming FtsA*-like (Fig. 7c). Although the molecular mechanism is unclear, this finding suggests that either in solution or on the membrane, the stimulatory effects of disrupting FtsA minirings on FtsZ lateral interactions can be potentially self-reinforcing.

Our results suggest a key role for FtsA minirings in aligning and stabilizing long FtsZ protofilaments on the membrane. The FtsZ protofilaments we observe on FtsA miniring carpets are considerably longer than those assembled in solution and notably aligned but unbundled. Therefore, as FtsA is required for membrane attachment of FtsZ, the results suggest that FtsA minirings keep FtsZ protofilaments separated and aligned. The μm-size swirls of FtsZ observed on monolayers seeded with FtsA[17] probably reflect a higher-order organization of the aligned FtsZ polymers on FtsA arrays that we report here, as the structures assembled by FtsA in their experiments were not reported. It should also be emphasized that FtsZ has a flexible protein linker between the polymerizing domain and the FtsA-interacting C-terminal peptide[53,54], and this linker needs to contain a minimum number of amino acid residues for function. We speculate that the lipid monolayer assembly assay and/or the staining method probably compress the linker. As a result, the apparent direct interaction between the main part of the FtsZ protofilament and two of the FtsA miniring subunits may not occur when the FtsZ linker is fully extended. However, it is not yet known whether or when the FtsZ linker is fully extended in the E. coli cytoplasm, or whether the linker can become more structured when bound to FtsA, thus reducing the distance between the two proteins.

Although a detailed mechanism for FtsA miniring-mediated FtsZ organization is not yet clear, it is reminiscent of models for how two B. subtilis membrane-associated FtsZ-binding proteins may organize FtsZ filaments at the membrane surface. In the first example, SepF of B. subtilis forms rings on the membrane that organize FtsZ filaments[55,56]. Like FtsA, SepF has an amphipathic membrane targeting helix, and can largely substitute for FtsA in B. subtilis[57]. Nevertheless, SepF seems to promote FtsZ protofilament bundling instead of inhibiting it like E. coli FtsA, and it is not yet known whether B. subtilis FtsA even forms minirings, and hence these remain open questions. The second example is B. subtilis EzrA, a spectrin-like membrane protein that binds to FtsZ and may antagonize bundling of FtsZ polymers at the membrane by forming arch-like structures[58].

Why might E. coli FtsA form minirings instead of straight actin-like polymers observed for other FtsA proteins? One intriguing possibility, consistent with prior genetic evidence, is that the closed minirings act as a molecular lock to keep FtsA and the divisome in its initial proto-ring stage until the divisome is ready to proceed[59] (Fig. 7a). In this model, overproduction of FtsA results in an excess of such closed or mostly closed FtsA

minirings that is toxic for the cell because the locked minirings block FtsZ protofilament bundling and consequently septal progression. The ability of FtsA* and FtsA*-like mutants to accelerate septum formation and bypass ZipA[14] indicates that these less geometrically constrained forms of FtsA may override the locking mechanism, in part to more efficiently recruit downstream divisome proteins[22]. Our results here with membrane-attached FtsA* showed that FtsZ protofilaments tethered to these nonring forms of FtsA assume more diverse orientations and superstructures, including bundles (Fig. 7b). Putting these ideas together, we propose that in cells, WT FtsA is converted from an initial miniring form to more monomeric, FtsA*-like nonring forms as the divisome matures (Fig. 7d). This conversion would switch WT FtsA from a constraining force on FtsZ protofilament orientation and bundling (as minirings) into a more densely packed carpet of flexible membrane tethers that release these constraints and allow more lateral interactions between FtsZ protofilaments. In this scenario, overproduction of WT FtsA blocks cell division by preventing those FtsZ lateral interactions, and can only be overcome by forcing increased FtsZ bundling by overproduction of FtsZ. We further propose that the conversion of FtsA minirings into smaller oligomers (or monomers) is reinforced in a positive feedback loop by the lateral interactions between FtsZ protofilaments themselves. This would have the combined effect of stabilizing FtsZ lateral interactions and blocking FtsA oligomerization, driving septum synthesis irreversibly forward. Increased FtsZ filament density is thought to be crucial for this constriction process[60].

This model is consistent with FtsA antagonism of FtsZ protofilament bundling we observe in vivo that we propose prevents this switch towards FtsZ bundling and thus septum formation. In addition to the potential FtsZ-mediated unlocking of FtsA minirings, one of many predictions of this model is that divisome proteins such as FtsN and FtsX, both of which can interact with FtsA and promote septum synthesis[24,25,61], do so by disassembling FtsA minirings. Now that membrane-bound FtsA structures can be monitored on lipid monolayers at high resolution, these questions and others, such as the effects of ZipA and Zap proteins on the bundling of FtsZ in the FtsA-membrane context, can be addressed using purified divisome proteins or protein domains to better approximate the complex cellular milieu. The recent finding that FtsZ polymers treadmill in vivo suggests that the FtsA miniring arrays and other oligomeric forms of FtsA are probably highly dynamic and form a moving track to tether treadmilling FtsZ protofilaments to the membrane. The high turnover of FtsA observed by photobleaching experiments in E. coli cells, as well as the higher turnover of FtsA*, are consistent with such rapid FtsA dynamics[14]. Other important future challenges will be to demonstrate the presence of FtsA minirings in E. coli cells (for example, by vitreous cryosectioning or thinning with a focused ion beam[62]), and to determine the molecular mechanisms by which FtsA minirings are disrupted by other factors.

## Methods

**Strains, plasmids and growth conditions.** For most genetic experiments, E. coli WM1074, a K-12 derivative of MG1655 (ilvG rpb-50 rph-1 ΔlacU169), a lab stock strain, was used as a WT background. Derivatives of TB28 (ref. 63) lacking zapA or both zapA and zapC have been previously described[64] and kindly provided by Professor Anuradha Janakiraman (CUNY). Another lab stock strain, DH5α, was used for gene cloning. BL21(DE3) (Novagen) was used for overproducing FtsZ and FtsZ* from pET11 plasmids, and C43, a derivative of BL21(DE3)[65] another lab stock strain, was used for overproduction of FtsA, FtsA* or FtsAΔC15 from plasmids pWM1260 (strain WM3857), pWM1609 (strain WM3260) or pWM4908 (strain WM4908), respectively. Cells were cultured in Luria-Bertani (LB) agar or broth at 30 °C unless otherwise indicated. LB medium was supplemented with ampicillin (50 μg ml$^{-1}$), kanamycin (50 μg ml$^{-1}$), chloramphenicol (15 μg ml$^{-1}$),

glucose (0.2%), sodium salicylate or IPTG when necessary. Standard protocols for molecular cloning, transformation and DNA analysis were used as previously described[66]. All plasmids used are listed in Supplementary Table 1.

**Cell fixation and immunofluorescence microscopy.** Overnight cell cultures were diluted 1:100 and grown to $OD_{600} = 0.2$ before being back diluted 1:4. Following the second dilution, cell were grown to $OD_{600} = 0.2$ and spotted on plates (for serial dilution plate assays) or diluted again 1:4 while inducing the expression of genes of interest. Cells were then maintained in mid-exponential phase ($OD_{600} = 0.4$-0.6) and collected 2 h after induction for microscopic analysis or immunoblotting. For visualization by differential interference contrast, cells were fixed with 1% formaldehyde. For immunofluorescence microscopy analysis, cells were fixed using one volume of methanol/acetic acid (4:1) mixture[67,68] followed by their preparation and imaging as previously described[69]. Briefly, fixed cells were added to wells of 15-well slides pretreated with 1% poly-L-lysine. After incubating for 2 min, the wells were aspirated, washed with phosphate-buffered saline (PBS) and 2 mg ml$^{-1}$ lysozyme was added and incubated for 4 min. Then, the cells were washed with PBS and blocked with PBS + 2% bovine serum albumin, followed by incubation with affinity-purified anti-FtsZ rabbit polyclonal antibodies (lab collection, 1:2,500 dilution)[70] for at least 1 h. After extensive washing with PBS, the wells were incubated at least 1 h in the dark with secondary goat anti-rabbit antibodies, conjugated to AlexaFluor488 (Thermo Fisher Scientific A-11034), at 1:200 dilution[71]. After washing with PBS, 10% glycerol in PBS was added to wells before placing a coverslip for storage and viewing.

**Western blot analysis.** Cell extracts were prepared, and proteins were separated by SDS–polyacrylamide gel electrophoresis and transferred as previously described[61]. Cellular levels of FtsZ or ZapA were measured on immunoblots by affinity purified rabbit anti-*E. coli* FtsZ (lab collection; 1:5,000 dilution)[70] or rabbit anti-*E. coli* ZapA antiserum (lab collection; 1:2,000 dilution). Rabbit anti-FLAG antibody (Sigma-Aldrich F7425) was used to measure FLAG-tagged FtsA levels (1:4,000 dilution). Goat anti-rabbit secondary antibodies conjugated to horseradish peroxidase (Sigma-Aldrich AQ132P) were used at 1:10,000 dilution. A Western Lightning ECL Pro Kit (Perkin-Elmer) was used for horseradish peroxidase detection and blots were developed using an ImageQuant LAS 4000 mini-image analyser.

**Protein purification.** FtsZ and FtsZ* were purified by precipitation with ammonium sulfate as previously described[71]. Purified FtsZ and FtsZ* were resuspended in storage buffer (50 mM Tris, pH 7.5, 250 mM KCl, 10 mM MgCl₂, 1 mM EDTA and 10% glycerol). GDP (0.05 mM) was added and protein aliquots were frozen in liquid nitrogen and stored at −80 °C.

His₆-tagged FtsA, FtsAΔC15 and FtsA* were purified using Talon metal affinity resin as previously described[72] with the following modifications. Cell pellets were washed in buffer (50 mM sodium phosphate pH 8.0, 300 mM NaCl) before centrifugation and storage at −80 °C. Pellets were then resuspended in purification buffer (50 mM Tris, pH 7.5, 250 mM KCl and 10 mM MgCl₂) with 5 mM imidazole, 1 mM phenylmethylsulfonyl fluoride and cOmplete EDTA free protease inhibitor tablets (Roche) added. Following incubation of lysate with ∼3 ml of resin pre-equilibrated in purification buffer, the resin was washed 2 × using 200 ml of purification buffer containing increasing concentrations of imidazole (5 and 20 mM). FtsA was then eluted with purification buffer containing 150 mM imidazole. FtsA-containing fractions, identified using Coomassie Plus Protein Assay reagent (Thermo Fisher Scientific), were pooled and dialysed in storage buffer. Then, 0.05 mM ADP was added before freezing aliquots in liquid nitrogen and storing at −80 °C. Protein concentrations were determined using the CB-X protein estimation assay (G-Biosciences).

**Liposome cosedimentation assay.** Liposomes were prepared using 99% *E. coli* total lipids (Avanti Polar Lipids Inc., Alabaster, AL, USA). Chloroform was evaporated from lipids using a nitrogen gas stream. Lipids were placed in a speed-vac for 30 min to make sure all chloroform was evaporated, and then were resuspended in buffer B (20 mM Tris, pH 7.5, 100 mM KCl, 25 mM potassium glutamate and 5 mM MgCl₂) using tip sonication for 20 s. Resuspended lipids were passed through an extruder with a 400 nm pore size (Avanti). FtsA proteins were diluted to 2 μM in buffer A (20 mM Tris, pH 7.5, 100 mM KCl, 25 mM potassium glutamate, 5 mM MgCl₂, 20% glycerol and 1 mM dithiothreitol) and then further diluted to 1 μM with buffer B either with or without liposomes added in a final reaction volume of 100 μl. ATP (2 mM final concentration) was added where indicated and samples were incubated at 37 °C for 2 h before ultracentrifugation at 60,000 r.p.m. in a TLA 100.3 rotor (Beckman). Supernatant fractions were removed and added to tubes with 20 μl of 5 × SDS buffer. Pellets were resuspended in 100 μl of buffer (50% buffer A and 50% buffer B) and 20 μl 5 × SDS buffer was added. Samples were vortexed and boiled before separation by SDS–polyacrylamide gel electrophoresis and Coomassie brilliant blue staining and destaining. Gels were scanned and analysed using ImageJ to determine the amount of protein in the supernatant and pellet for each sample[73].

**Lipid monolayer assay.** The two-dimensional lipid monolayers were prepared using *E. coli* polar lipids (Avanti) as previously described[30]. Briefly, 0.2 μg of lipids (per well) were floated on storage buffer (lacking EDTA) in a custom-made Teflon block and placed for 1 h in a humid chamber to let the chloroform evaporate. Electron microscopy grids were placed on the monolayers, followed by sequential additions and incubations of 0.1–1 μM FtsA (40 min), 4 mM ATP (20 min), 5 μM FtsZ (5 min) and 4 mM GTP (20 min) in a total volume of 90 μl. The grids were removed and negatively stained with 1% uranyl acetate as previously described[28]. Grids were inspected and photographed with JEOL 1400 Transmission Electron Microscope coupled with a Gatan Orius CCD camera. Protofilament spacing was measured with boxes or lines using the Plot Profile tool in ImageJ.

**Tomography data collection and reconstruction.** Negatively stained samples were imaged with a 300 kV electron microscope (FEI Polara) equipped with a field emission gun and a direct detection device (Gatan K2 Summit). The tomographic package SerialEM[74] was utilized to collect a single-axis tilt series at ∼6 μm defocus with cumulative doses of ∼200 e⁻ Å⁻². The data were sampled at a pixel size of 4.5 Å. For each data set, over 41 image stacks were collected in a range from −60° to +60°, with a 3° increment. Each stack contained ∼10 images that were first aligned using Motioncorr[75] and were assembled into drift-corrected stacks by TOMOAUTO[76]. The drift-corrected stacks were aligned and reconstructed by using marker-free alignment[77]. In total, 23 tomograms (3600 × 3600 × 120 pixels) were generated for detailed structural analysis of the FtsA–FtsZ interactions.

**Subtomogram analysis.** We first identified the minirings manually in one tomogram. Approximately 200 rings were extracted and averaged to generate an initial model that is essentially a donut-shaped structure. It was then used as a template to automatically identify particles from 23 tomograms by using template matching implemented in PyTom[78]. A total of 18,400 ring-like structures (128 × 128 × 64 pixels) were extracted for subtomogram analysis by using tomographic package I3 (ref. 79). Two distinct class averages emerged after multiple cycles of alignment and classification. One class average shows the rings packed hexagonally. Another class average has the rings attached to one filament-like density.

**Molecular modelling.** The class average of hexagonally packed rings without a filament density was used for the fitting of the crystal structure of *S. aureus* FtsA in complex with ATP (PDB: 3WQU)[48] using UCSF Chimera[80]. The A and B chains of one copy were manually fit into the density and we optimized its placement by localized rigid-body fitting. Further copies were docked by first aligning to the initial fitting and then rotating the copy in increments of 60° about the symmetry axis of the density map (Supplementary Movie 2). The crystal structure of *Thermotoga maritima* FtsA in complex with a segment of FtsZ (PDB: 4A2A)[32] was aligned with the docked copies using the matchmaker function in Chimera. We isolated the filament density used in the modelling by calculating the difference map between the two class averages.

**Data availability.** The authors declare that all data supporting the findings of this study are available within the article, Supplementary Information or from the authors on request.

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

## Acknowledgements

We thank Anuradha Janakiraman (CUNY) for strains, Daniel Haeusser for the pKG110-FtsZ$_{E93R}$ plasmid, Miguel Vicente and Anabel Rico (CNB-CSIC-Madrid) for advice and for providing us with the custom-made Teflon block for making lipid monolayers and Daniel Vega Mendoza for helpful discussions. M.K. dedicates this work to his parents, Stanislaw (1957-2017) and Dorota. This work was supported by NIH award GM61074 to W.M. and GM107629 to J.L.

## Author contributions

All of the authors contributed to the research design, protein and lipid purification and technical troubleshooting. V.W.R., K.S. and M.K. performed the electron microscopy experiments; M.K. did the *in vivo* experiments; J.L. and D.M. performed the tomography and modelling and assembled relevant figures and movies; M.K. and W.M. wrote the manuscript.

## Additional information

**Competing interests:** The authors declare no competing financial interests.

**Publisher's note**: 

