## [Peer Review File · Nature Communications]

Reviewers' comments:

Reviewer #1 (Remarks to the Author):

This paper presents two rather separate studies, and attempts to connect them as related to FtsZ protofilament bundling. The first study investigates in vivo effects of mutants of FtsZ and A, and overexpression and deletion of ZapA and C, accessory proteins that have been attributed to bundling FtsZ. This study is reasonably consistent, but I am not convinced by the interpretation where the key role is. The second study is in vitro EM of FtsZ and A interacting on lipid monolayers. The discovery that FtsA makes minirings on the lipid monolayer is very novel and flashy, but fitting this into a mechanism is speculative and not convincing.

An overall problem for me is the central focus on "FtsZ protofilament bundling." I don't see this as a single, defined phenomenon, but rather a term applied loosely to some very different mechanisms. Crowding agents (not studied here) produce large round bundles; CTV bundles (not studied here) only form in very low salt; ZapA produces lacy sheets, probably 2-D, but their structure in the literature is still confusing; Ca produces irregular thin bundles; E93R seems to make large arrays that may be 2-D sheets (only one published image); L169R, which is presented here as the archetype bundler, seems to make primarily protofilament pairs, which I would not even classify as bundles. The in vivo mutational study makes a game effort to relate all the effects to enhancing or inhibiting bundling. There is, however, no mechanism for bundling, and the range of structures described as bundles questions whether there is a single mechanism.

Especially important to my thinking, there is no suggestion of a mechanism by which FtsA might inhibit bundling. FtsA has only one known contact with FtsZ, the Ct peptide. The in vitro study attempts to make an association that lipid monolayers with A* show more bundling (actually protofilament pairs of L169R) than A, but there is no mechanism. I suspect it may be the lower concentration of A* than A on the monolayers (see below).

There is also a speculation that the minirings of A somehow guide the Z protofilaments into parallel arrays where they are roughly parallel but not in contact. Fig. 6g,h shows the Z protofilament contacting A precisely on two subunits just above the midpoint of the miniring. This seems to ignore that the linker from Z to A is a flexible peptide of 50 aa's. Any given Z could contact any A within 10-17 nm, and the A could be in any orientation. I see nothing in the images that suggests a preferred binding site on the miniring. I would suggest that the parallel arrangement may come from remodeling and growth of protofilaments on the FtsZ monolayer – once a protofilament has established itself, a new protofilament nearby may be inhibited to cross it (attachments to A would keep it in the plane) and so it could only grow parallel. The timing of the EM specimens was not specified, but I would imagine that they were made after many rounds of growth and remodeling. Assembly that started out random evolves into parallel arrays as the FtsZ protofilaments are inhibited from crossing.

Overall I think there are a number of very interesting observations here, but I am not convinced by the interpretations.

Detailed concerns and questions:

Methods for IFM "cells were fixed using one volume of methanol:acetic acid (4:1)mixture followed by preparation and imaging as described." The ref "described" says "as previously described." Please give the precise conditions used in this study.

The more diffuse width of the Z rings induced by extra FtsZ is potentially important but perhaps not yet convincing. Fig. 2b looks convincing, but S2 less so. The major difference in S2 seems to be that wt Z rings are overall wider than in ZapA/C. Unless this can be verified with statistics I suggest removing it. The filamentation phenotype however is sound.

p. 8 l. 8: "antagonizes its assembly into bundles." I think it is Z rings that are antagonized. The authors may consider the Z ring a form of bundle, but I think it would be better to be conservative. Lines 9-10 continue the interpretation that "FtsA inhibits the assembly of FtsZ into

bundles." It actually inhibits the Z ring as seen by IFM.

V208A is presented as another bundling mutant. However I haven't found any evidence for its bundling. It antagonized the kil inhibition like L169R, and the authors seem to have formed the conclusion that this is due to bundling. But they really need to repeat the EM, pelleting, and GTPase assays that established enhanced bundling for L169R. E93R was pretty convincing at bundling in Jaiswal, but they might want to repeat this also.

R174D was originally described to eliminate bundling in Ca, but Moore...Erickson 2016 have recently presented evidence that it bundles just like wt. It is apparently non-functional and toxic, but it should not be relevant to a bundling hypothesis.

Fig. S5 seems very important. It makes two points. (a) That FtsA must be able to bind the membrane to be toxic in the DZapA/C background, and (b) that FtsA missing domain 1C or defective in ATPase are still fully active. It's not really clear how this fits the bundling hypothesis, but it does seem a potentially important observation.

The FtsA minirings are pretty spectacular and surprising. Nothing like them has been seen before. A similar assembly of TmFtsA on lipid monolayers showed relatively straight pfs, often in pairs. CryoEM of E. coli identified filaments of FtsA running parallel to FtsZ but closer to the membrane. There was no hint of anything running perpendicular to FtsZ, but this might be a limitation of the cryoEM in whole E. coli cytoplasm.

I doubt that FtsA minirings are physiologically or structurally important in E. coli for a number of reasons. (a) There are only 1,000 FtsA molecules in the cell. That could make 83 minirings at most, which could make a continuous row 2,100 nm long, less than the 3,000 circumference. The Discussion suggested that the minirings might be in ~6 clusters, but then we need something to bridge the clusters to keep the Z ring intact. (b) A major difference between the cell and the lipid monolayer: in the cell there are only 1,000 FtsA, but in vitro there are an infinite number of soluble FtsA available for binding. So if the binding affinity is high enough the monolayer in vitro can form a continuous layer, whereas in vivo the 1,000 A will only be able to cover a small fraction of the available membrane. (c) If the full miniring is not physiologically relevant, it does suggest a more limited interpretation. Apparently EcFtsZ likes to make curved pfs, and these close to make minirings when the concentration is sufficiently high on the monolayer.

From the tomogram reconstruction Fig. 6g I calculate a diameter of 23.6 nm for the mid-point of the ring (based on the 10 nm scale). This gives a spacing of 6.2 nm. This is much larger than the 4.8 nm spacing in the Sendia xtal structure, and the ~5 nm spacing of actins and most homologs. Magnifications should be checked and this discrepancy addressed.

It is difficult to imagine that the FtsZ filaments could be organized by the circles and arcs of FtsA. The FtsA-binding peptide of FtsZ is on the end of a flexible linker that can likely reach 10+ nm in any direction, as well as orient the peptide to match the orientation any nearby FtsA.

I am not convinced that lipid monolayer structures argue that FtsA inhibits FtsZ bundling, and that FtsA* is less effective. First, the EM images are anecdotes, and even then not that convincing. 7f-h shows FtsZ pf pairs, but not the larger bundles. Strikingly, one does not see any FtsA minirings in 7f-h. The authors note this lack and suggest "perhaps because of the FtsZ* polymer mass covering them." The discussion ignores the possibility that there may be binding reactions taking place in solution that modulate the concentrations available for binding the lipid.

But another important consideration is that the difference in A and A* may be simply the amount on the monolayer. The binding to the monolayer is probably a cooperative event, where the weaker A*-A* interaction would mean less bound than the A-A. This might be approached semi-quantitatively by fluorescent Ab labeling. At the very least this idea should be pursued by

increasing the concentration of A*. It might even form complete minirings if its surface concentration were raised to match that of A.

A related question is the order of addition. Fig. 7e caption says + "0.5 μM A* + 5 μM Z*" while p.14 l.8-9 says "we added 0.5 μM A* to the monolayer and then added 5 μM Z*". This may make a big difference, whether the A and Z are first equilibrated in solution, or A* is first equilibrated with the monolayer.

Reviewer #2 (Remarks to the Author):

The paper provides new insights into the oligomerization of the actin homolog FtsA protein and its implication for the bundling of FtsZ filaments. It combines *in vivo* data on the cell viability and FtsZ ring morphology at different genetic backgrounds with the biochemical reconstitution of the purified proteins on a lipid monolayer. The latter allows for a high resolution characterization of FtsA oligomeric state by EM. Together the authors suggest a model by which a higher-ordered state of FtsA (membrane-bound minirings) antagonizes bundling of FtsZ filaments.

The manuscript is organized in two parts: the first part is dedicated to *in vivo* experiments where the phenotypes and viability of the strains overexpressing FtsA, FtsZ and Zap(s) proteins are characterized. These experiments are based on previous observations that although overexpression of FtsA is lethal, its toxicity can be suppressed by either co-overexpression of FtsZ [Dai & Lutkenhaus 1992, Dewar et al 1992] or by the presence of hyperbundled FtsZ (FtsZ*) [Haeusser et al 2015]. The authors substantiate these findings by showing that the negative influence of FtsA on FtsZ bundles is counterbalanced by the bundling activity of ZapA and ZapC. Accordingly, the toxic effect of overexpressed ZapA can be compensated for by co-overexpression of FtsA.

Overall, the results shown in this part are convincing, however it is disappointing that the paper does not include more recent imaging techniques or image analysis to better quantify the influence of ZapA, ZapC and FtsA on FtsZ filament bundling. Apart from the not-convincing model (see below), I think this is biggest weakness of the paper. I think it would great benefit from one of the three following approaches:

Super-resolution microscopy of FtsZ filaments in wt and mutant cells. This is pretty much standard these days and as ZapA, ZapC and modest increase of intracellular FtsA seem to have only subtle effects on FtsZ ring architecture (see Fig. S2), their influence can be visualized much more convincingly at higher spatial resolution.

live-cell imaging. It is not clear to me why this is not standard nowadays. A number of recent high-impact papers have demonstrated very beautifully that by just look at the dynamics of proteins *in vivo* a much better understanding of intracellular organization can be achieved (see for example Garner et al, Domínguez-Escobar et al, van Teeffelen et al, 2011 or Bisson Filho et al, Yang et al, 2016 (bioRxiv). This will also be true for this paper.

better statistics/image analysis. Fig. S2 is very disappointing, do these lines represent fluorescence intensity profiles along individual cells? The intensity distributions of FtsZ rings greatly vary between different cells and during the cell cycle. Accordingly, such a representation is rather meaningless. The authors should normalize the intensity profile for different cell lengths, average the intensity profile over many cells and normalize the intensity profiles. The authors might find this tool useful: <https://sils.fnwi.uva.nl/bcb/objectj/examples/Coli-Inspector/Coli-Inspector-MD/coli-inspector.html>

In the second part, perform biochemical experiments and electron microscopy to elucidate a

potential molecular mechanism by which FtsA prevents FtsZ bundling.

By using purified proteins and lipid monolayers, the authors found that FtsA is able to organize into minirings on a membrane. This is first visualization of membrane-bound filaments of *E. coli* FtsA and the formation of FtsA minirings is in contrast to previous observations of *T. maritima* FtsA forming long straight filaments [Szwedziak et al 2012]. When polymerized FtsZ was added, FtsZ filaments were found to be recruited to the FtsA covered membrane, forming long, non-bundled filaments. This data is beautiful and will certainly attract the attention of the field.

However, the second main problem of the paper is that the authors fail to convincingly interpret their observations and to propose a unifying model. Especially in light of recent findings of dynamic treadmilling of FtsZ and FtsA in vivo [Bisson Filho et al, Yang et al, 2016 (bioRxiv)] and in vitro [Loose & Mitchison 2014], the authors need to do a better job in putting their observations in perspective.

For example, the authors appear to suggest that FtsA disrupts FtsZ bundles by assembling into minirings, which keep FtsZ filaments at a certain distance. This fragmentation of FtsZ bundles is important to perform cell division. However, as suggested in the last figure, at in vivo conditions, FtsA might not form rings but arcs or shorter oligomers. Accordingly, it must disrupt FtsZ bundles using a different mechanism than via FtsA minirings. Instead these minirings might represent the toxic FtsA species able to prevent cell division.

In summary, this paper further substantiates previous observations that FtsA has the ability to disrupt FtsZ filaments or FtsZ filament bundling. The EM data sheds light on *E. coli* FtsA structures. However, the paper fails to propose a convincing model that could help understand how FtsA, FtsZ and Zaps interact to coordinate cell division. For publication, major revisions are needed.

In addition, I do have a number of other questions that I hope the authors can address:

1st part, in vivo experiments

page 7, line 5: the picture of the gel appears to have been stitched together from multiple experiments. The authors should either make it obvious that it originates from different gels or repeat the experiment with all samples on one gel.

page 7, line 13: Evidence in lines 13-22 is very weak. Fig. S2: Do the lines correspond to three individual cells? FtsZ ring widths should be plotted as a function of the cell length and averaged over many cells to make a strong statement. For example, if Zaps are bundling proteins it does not make much sense that the rings appear to be tighter in those cell lacking Zaps.

page 9, line 17: Does FtsA prevent FtsZ polymerization and bundling in solution or on a membrane? While the present manuscript suggests that FtsA would only be active when both proteins are in contact with each other on the membrane, in a previous paper [Beuria et al 2009], the lab of Margolin showed that FtsA shortens FtsZ filaments in solution.

page 9, line 24: "FtsA Δ 1C, lacking the 1C domain important for self-interaction and recruitment of late division proteins, ..., inhibited growth when induced, like WT FtsA". This suggests that FtsA does not need to form minirings to exert its antagonistic effects on FtsZ polymer bundling.

2nd part, in vitro EM

For their biochemical experiments, the authors use N-terminal 6xHis-tagged FtsA expressed from pET28, which also contains a thrombin site [Geissler et al 2003] It is well known that the Polyhistidine can alter the solubility and assembly properties of protein polymers [Petek & Mullins 2014], therefore it is important that the authors demonstrate that FtsA still forms membrane-bound minirings upon removal of the 6xHis-tag.

Expression of FtsA from pDSW210F-ftsA at 500 μ M IPTG is toxic. How could the authors purify FtsA from pET28 at 1 mM IPTG?

Why their FtsZ filament are so long, more than 1 μ m in length? This is much longer than observed previously. If FtsA is increasing FtsZ dynamics and decreasing FtsZ stability [Beuria et al 2009,

Loose & Mitchison 2014], should the filaments not be much shorter than that? Would shorter filaments bind to FtsA with a random orientation?

The density of FtsA on the membrane depends on the absolute amount of protein present in solution i.e. the surface to volume ratio at a given concentration. What was the buffer volume and the surface area in the experiment?

Is FtsA oligomerization important for membrane binding? If yes, one should see a highly sigmoidal curve in FtsA-titration experiments (amount of membrane-bound FtsA vs. FtsA concentration).

On the EM micrographs it looks as if there was much more FtsA bound to the membrane than FtsZ. Given that the authors say they used physiological protein concentration ratios, could the authors comment on that apparent discrepancy?

The authors used a rather long incubation times for FtsA and ATP (1 hour total), especially compared to Loose & Mitchison 2016, where the dynamic behavior of the proteins were found within at earlier time points. Is the formation of FtsA time-dependent? Does it depend on the order by which the proteins were added?

How does the diameter of the ring, and therefore the distance between two FtsZ binding sites, compare to the distance between the corresponding FtsZ monomers?

Discussion

Fig. 8. can be improved. For example the authors could include an illustration of how FtsA minirings might be organized in the cell (see page 15, line 22-page 16, line 2).

Reviewer #3 (Remarks to the Author):

In this paper, the authors characterize the role of FtsA and FtsZ in *E. coli* divisome assembly. The authors present genetic evidence using a number of mutants that promote or inhibit FtsZ bundling to support the hypothesis that FtsA antagonizes FtsZ bundling in living cells. Remarkably, in vitro structural studies using electron microscopy of purified components show that physiologic concentrations of FtsA assemble into minirings on lipid monolayers and that these rings guide the assembly and organization of FtsZ protofilaments and prevent their bundling. The authors then focus on a hypomorphic mutant of FtsA that accelerates septum formation and bypasses the requirement of ZipA. In vitro results show that bound FtsA* does not form minirings but assembles into small oligomers that promote assembly of disorganized and bundled FtsZ protofilaments. Taken together, the in vivo and in vitro evidence supports the hypothesis that FtsA acts to antagonize FtsZ bundling and that the oligomeric state of FtsA is important to influence the organization FtsZ. The authors present a working model where the oligomeric state of FtsA serves as a molecular lock in early stages of protoring formation to keep FtsZ unbundled until proteins are recruited during maturation of the divisome. FtsA is then converted to non-ring forms (presumably by other divisome proteins) which then subsequently releases constraints on FtsZ orientation and bundling.

This is an important and interesting paper that uses a comprehensive approach with genetic, biochemical and structural methods to establish the roles of FtsA and FtsZ in bacterial cell division machinery. Admittedly, this mechanism may not be universal to all bacteria. The results complement and extend previous light microscope observations by Loose and Mitchison on the importance of FtsA in tethering FtsZ to lipid monolayers. The paper is well written and clearly lays a path of important genetic and structural results to support their hypothesis that FtsA performs a more regulatory role and acts to antagonize the bundling of FtsZ protofilaments. However, there are a number of major and minor points that need to be addressed prior to publication.

Major:

The authors present beautiful tomography and 3D volume averaging of FtsA and FtsZ organization in vitro. The resulting tomograms clearly show minirings of FtsA bound to lipid monolayers. Each FtsA miniring is formed from twelve subunits. Unbundled FtsZ protofilaments bind to two of the twelve subunits of the membrane-distal region of the FtsA ring. However, the 2D images of the negative stained samples of FtsA* bound to lipid bilayers were less convincing. The study would benefit from tomography of FtsA* with FtsZ to visualize the underlying FtsA* fragments more clearly and to understand the interaction of FtsZ with these short oligomers. For example, does FtsZ still maintain binding to two FtsA* subunits? This would bring more detailed and realistic information to support the model presented in Figure 8b.

Use of negative stain samples for high resolution averaging and docking of crystal structure is not typical due to the fact that these samples are dehydrated and heavily stained. The authors correctly point this fact out on page 12 (lines 19-1) where they point out that higher resolution structures would be needed to confirm the model. The limitations using negative stain samples is also eluded to in the discussion on page 16 (lines 19-21) where the authors point out that the staining method could lead to compression of flexible linkers in the protein. The use of negative stain samples in this study was appropriate in that it provided a method to illustrate the overall organization of FtsA and FtsZ under various conditions. A statement in the discussion that future studies using cryoEM of vitrified samples will be important to obtain higher resolution structures to overcome the limitations of the negative stain technique.

The organization of FtsA minirings in vitro presented in this paper was remarkable, but as the authors point out there is no direct evidence that minirings are present in cells. The discussion could be developed further by proposing that future studies using cryoET may be useful for visualizing the cell division machinery in vivo, similar to what was done with *Caulobacter crescentus* (Li et al., 2007). Although *E. coli* is too large for whole mount cryoET, methods such as vitreous cryosectioning (CEMOVIS) or cryo FIBSEM may offer important avenues for identifying these structures in a near-native state.

Minor:

Discussion: page 16, line 7: FtsA should be FtsA*

Figure 4. I would expect Figures 4c and 4f to look more similar. To my eye Figure 4f looks more like Figure 4d. Some comment about variability of the preparation should be mentioned.

Figure 5. Figure 5d. The panel shows 0.5 μ m FtsA + 5 μ M FtsZ on a grid without a lipid monolayer. The text says 'only FtsZ filaments were detected'. Some statement about the absence of FtsA rings should be added. There is a lot of background material that could be unassembled FtsA and there does appear to be more filaments than in the control (Figure 5a) where FtsZ is added to lipid bilayer alone.

Movie S2 was mentioned in the Methods section but not mentioned in the Results section of the text. Perhaps it could go on page 12 line 19: (Figure 6h; movie S2). The movie could also be mentioned in the legend of Figure 6h.

Figure 7. It would be useful to have arrows pointing to arcs and short curved protofilaments or even an inset at higher mag in figure 7b. It was difficult to see these structures at the magnification of the image.

Figure 7f-h. Page 13 (line 25) and page 14 (line 1) the authors point out that FtsA rings were difficult to detect because of the large FtsZ* mass covering them and I agree. Tomography on these samples should reveal the underlying structure of the FtsA beneath the thick polymer mass of FtsZ*.

Figures S6 and S7 should have scale bars on the images. FigS6 compares periodicity from a tomographic slice in the top panel (a) with 2D projection of the mutant in (c).

Technical details in the methods on Page 21 lines 13-22. "Tomography data collection and reconstruction"

-The pixel size at which the data were acquired should be explicitly mentioned. This is important for the reader to know the resolution of the tomograms and subvolume averages. Since SerialEM was used to acquire the data, this information will be in the image header.

-It is unusual to collect images of negative stained samples with such a large defocus (-6 μ m). This is typically used for unstained samples in cryoEM. Was CTF correction used on these data? The authors incorrectly cite a program "TOMOAUTO" as Kremer et al. and Xiong et al. The Kremer et al. paper references the IMOD software package for alignment and reconstruction and the Xiong et al. references methods for CTF correction.

-Similarly, on page 21 line 21 there is a (5) at the end of the sentence about alignment and reconstruction. Perhaps this is where the Kremer and Xiong references should be correctly cited.

Page 17, line 23 should read "both of which"

We thank the reviewers for taking the time to read the manuscript thoroughly and make
constructive comments for its improvement. Our corresponding responses in italics are
below each reviewer comment. Please note that we removed the old Fig. 3, so the old
Figs. 4-8 are now Figs. 3-7. The responses refer to the new numbering system.

Reviewer #1 (Remarks to the Author):

This paper presents two rather separate studies, and attempts to connect them as related
to FtsZ protofilament bundling. The first study investigates in vivo effects of mutants of
FtsZ and A, and overexpression and deletion of ZapA and C, accessory proteins that
have been attributed to bundling FtsZ. This study is reasonably consistent, but I am not
convinced by the interpretation where the key role is. The second study is in vitro EM
of FtsZ and A interacting on lipid monolayers. The discovery that FtsA makes minirings
on the lipid monolayer is very novel and flashy, but fitting this into a mechanism is
speculative and not convincing.

*We appreciate the reviewer's enthusiasm about the novelty of our findings, which,*
*along with the first explanation for how FtsA assembly state can regulate FtsZ assembly*
*state, we believe justifies its publication in Nature Communications. Our in vivo and in*
*vitro studies complement each other in a way that either one alone could not. The*
*genetic/cytological evidence is used to promote the key hypothesis that FtsA*
*antagonizes lateral interactions between FtsZ protofilaments; although the evidence is*
*indirect (like most genetic/cytological evidence), its strength is that it happens in cells.*
*The EM study then tests this hypothesis using a purified system, which provides strong*
*complementary support for the in vivo studies.*

*Together, the in vivo and in vitro observations strongly suggest that FtsA minirings*
*inhibit direct lateral interactions among FtsZ protofilaments, whereas non-ring FtsA,*
*perhaps because of its higher packing density on the membrane, seems to promote*
*strong lateral interactions among FtsZ protofilaments. We agree with the reviewer that*
*the molecular basis of these different FtsZ-FtsA interactions is currently unclear and*
*needs further investigation using higher resolution methods that are beyond the scope*
*of this study. But we believe that our model explains the data very well, despite needing*
*refinement with further studies, as with any model. Our proposed general mechanism*
*of FtsA miniring unlocking, accompanied by FtsZ protofilament bundling, explains*
*many genetic studies of E. coli cell division regulation for the first time, including the*
*ability of FtsA*-like gain of function mutants to bypass ZipA and the existence of*
*separate stages of divisome maturation. Because our model involves the two main*
*conserved actin and tubulin proteins that regulate bacterial cell division, we think that*
*it will serve as a valuable conceptual foundation for further understanding of the*
*regulation of cell division in E. coli and potentially in other species. We therefore think*
*that combining the in vivo and in vitro studies makes a stronger case.*

*To address the reviewer's concern about the mechanism not being convincing, we*
*modified the model in Fig. 7 (was Fig. 8) as well as the explanatory text to make a*
*better case for why our results fit quite nicely with a mechanism that could explain how*
*cell division is regulated by FtsZ/FtsA higher-order assembly state.*

An overall problem for me is the central focus on "FtsZ protofilament bundling." I don't
see this as a single, defined phenomenon, but rather a term applied loosely to some very

different mechanisms. Crowding agents (not studied here) produce large round bundles;
CTV bundles (not studied here) only form in very low salt; ZapA produces lacy sheets,
probably 2-D, but their structure in the literature is still confusing; Ca produces irregular
thin bundles; E93R seems to make large arrays that may be 2-D sheets (only one
published image); L169R, which is presented here as the archetype bundler, seems to
make primarily protofilament pairs, which I would not even classify as bundles.

*We agree with the reviewer that there seem to be multiple FtsZ higher-assembly states*
*and multiple mechanisms for attaining those states, and that we may only be examining*
*a subset in this study. We were aware of this potential diversity of mechanisms and that*
*is why we purposely used the general term “bundling” to avoid overinterpreting any*
*particular type of assembly state (and to provide a simpler way to describe strong*
*lateral interactions between protofilaments). The term “bundling” has been used*
*extensively in many recent papers (e.g. Loose & Mitchison) when referring to lateral*
*interactions between FtsZ pfs. However, because we did not define “bundling” in the*
*original submission, we have now defined it as any type of higher oligomeric state*
*involving lateral intermolecular interactions between pfs. We also changed the wording*
*of the title from “protofilament bundling” to “lateral interactions” to avoid any*
*ambiguity.*

*We suggest that the mainly double protofilaments formed by FtsZ* are simply part of*
*the bundling spectrum (perhaps this is the nucleus that gives rise to most other*
*bundles). Of course, FtsZ* has the potential to form larger bundles than WT FtsZ*
*under certain conditions (e.g. in the presence of FtsA*, see Fig. 6), indicating that*
*FtsZ* is not limited to double pfs and that double pfs can easily form higher order*
*bundled pfs. Thus we think that this comes down to semantics, and given the diversity*
*of mechanisms and their uncertainty that the reviewer describes, we think that pf pairs*
*still fall under the general term “protofilament bundling” as we now define it.*

The in vivo mutational study makes a game effort to relate all the effects to enhancing
or inhibiting bundling. There is, however, no mechanism for bundling, and the range of
structures described as bundles questions whether there is a single mechanism.

*We agree with the reviewer that there are likely multiple mechanisms for bundling, but*
*we present the first evidence, based on genetic inferences, that FtsA can antagonize*
*some aspects of bundling in the cell. The aim of our study was not to characterize*
*different FtsZ bundling states or define a bundling mechanism per se, but to show that*
*FtsA can influence FtsZ higher order assembly on the membrane. Because of our*
*findings, perhaps FtsA can be used as a tool to help dissect these potentially distinct*
*FtsZ bundling mechanisms in the future, particularly if FtsA inhibits some mechanisms*
*but not others.*

Especially important to my thinking, there is no suggestion of a mechanism by which
FtsA might inhibit bundling. FtsA has only one known contact with FtsZ, the Ct
peptide. The in vitro study attempts to make an association that lipid monolayers with
A* show more bundling (actually protofilament pairs of L169R) than A, but there is no
mechanism.

*We assume the reviewer meant to say “(actually protofilament pairs of FtsZ that*
*resemble those made by L169R)”, because one of the key points in the manuscript (and*
*a crucial underpinning of our model) is that lipid monolayers with FtsA* induce*
*considerable lateral association of WT FtsZ protofilaments. This is key because*
*minirings of WT FtsA rarely permit FtsZ protofilament bundling/lateral interactions*
*under our conditions, thus indicating that breaking the FtsA minirings may be an*
*important factor in switching non-bundled FtsZ protofilaments into bundled ones. In*
*complete support of this model, FtsZ* (L169R), which normally forms mainly*
*protofilament pairs with FtsA, is stimulated to form huge bundles or sheets on the lipid*
*monolayers in the presence of FtsA*. This indicates that the effects of the two*
*mutations are additive, and are completely consistent with the deleterious effects in vivo*
*that were ascribed to too much bundling in Haeusser et al. 2015.*

*In addition to providing a general mechanism for changing FtsZ assembly state through*
*changes in FtsA assembly state, the data argue that there is an abundance of FtsA**
*bound to the lipid monolayers to induce FtsZ and FtsZ* bundling (see next comment*
*below). Therefore, we disagree with the reviewer that “there is no mechanism”, at*
*least at the resolution proposed. As mentioned above, we clearly suggest that the FtsA*
*minirings inhibit FtsZ bundling, whereas FtsA in non-ring structures permit bundling.*
*The reviewer is correct that we do not yet have a mechanism at the level of structure.*
*However, this must await a future study that can generate higher resolution data. We*
*think that the overall model of FtsA oligomeric state affecting FtsZ lateral interactions*
*is sufficiently novel and significant for the cell division field to stand on its own, and*
*like any model in molecular biology, can be further refined and tested with additional*
*investigation.*

I suspect it may be the lower concentration of A* than A on the monolayers (see
below).

*The reviewer suggests that FtsA* may be at a lower concentration on the monolayers*
*compared with a similar loading of FtsA because of its deficiency in self-interaction,*
*and this lower concentration could be a reason for why FtsZ is more bundled with*
*FtsA* than with FtsA. This would make a lot of sense if FtsA binds to the membrane*
*cooperatively. However, when 0.5 μ M FtsA* is added to the monolayers, curved*
*oligomers are very abundant in many fields (Fig. 6b), making it unlikely that FtsA* has*
*a significant membrane binding defect compared with WT FtsA. This concentration of*
*FtsA* promotes strong FtsZ bundling (Fig. 6d), whereas 0.5 μ M or 0.1 μ M WT FtsA,*
*the latter which forms fairly sparse FtsA rings and arcs on the monolayer, fail to*
*promote much FtsZ bundling. This argues that there is plenty of FtsA* on the*
*monolayers. There is additional compelling evidence from genetic assays and liposome*
*cosedimentation (shown later in our responses to reviewer 1 and in the supplementary*
*material) that also argues against a membrane binding defect of FtsA* relative to FtsA.*

There is also a speculation that the minirings of A somehow guide the Z protofilaments
into parallel arrays where they are roughly parallel but not in contact. Fig. 6g,h shows
the Z protofilament contacting A precisely on two subunits just above the midpoint of
the miniring. This seems to ignore that the linker from Z to A is a flexible peptide of 50
aa's. Any given Z could contact any A within 10-17 nm, and the A could be in any
orientation. I see nothing in the images that suggests a preferred binding site on the
miniring.

*We emphasize that the parallel arrays of separated FtsZ protofilaments on FtsA*
*minirings is not speculation as stated by the reviewer, but reproducible EM data. The*
*reviewer is absolutely correct about the FtsZ flexible linker, and that is why we pointed*
*out on p. 16 of the original manuscript the importance of the linker and its possible*
*compression during assembly and/or staining. However, because the images in Fig. 6g-*
*h are averaged, it strongly hints at a preferred binding site on the miniring, at least in*
*the highly structured miniring arrays. In these averaged images, FtsZ filaments almost*
*never interact with the center of the miniring (i.e., the 9:00 and 3:00 subunits on the*
*clock face) but instead off center, with the 10:00 and 2:00 subunits. We did not*
*emphasize this in the manuscript because at this point we do not have sufficiently high*
*resolution for the structures to propose a molecular mechanism based on the atomic*
*structures. Furthermore, there are many cases of FtsZ filaments bound to the*
*membrane by FtsA minirings that are not in such highly structured arrays. However,*
*the overall ultrastructure of the oligomers suggests a general mechanism for how FtsZ*
*protofilament orientation could be nucleated and/or constrained by an FtsA miniring*
*array.*

*Moreover, it is possible that binding of the FtsZ flexible linker to FtsA induces an*
*ordered state, with FtsA acting in a chaperone-like role. Given the evolutionary*
*relationship between FtsA and HSP70, as well as some examples of intrinsically*
*disordered proteins that become structured after binding to other proteins, this*
*possibility cannot be ruled out. Therefore, the assumption that the linker can extend*
*many nm may not be correct in the context of membrane-bound FtsA and FtsZ. Along*
*these lines, the reviewer suggests that the FtsZ flexible linker can extend 10-17 nm; yet,*
*in the discussion of the Gardner et al. paper about the E. coli FtsZ flexible linker, the*
*authors propose a 5-8 nm distance for the FtsZ linker based on its calculated contour*
*length. Even if the FtsZ linker remains disordered upon binding to FtsA, this distance is*
*considerably shorter than the 20 nm diameter of FtsA minirings, making it more likely*
*that the miniring geometry can effectively constrain the position of FtsZ monomers that*
*it contacts. Of course, in the cell there are proteins such as ZapB than may exert a*
*significant inward force on FtsZ that could fully extend the linker. All of this is quite*
*hypothetical at this point, so as with any in vitro system we must glean what we can.*

I would suggest that the parallel arrangement may come from remodeling and growth
of protofilaments on the FtsA monolayer – once a protofilament has established itself, a
new protofilament nearby may be inhibited to cross it (attachments to A would keep it
in the plane) and so it could only grow parallel. The timing of the EM specimens was
not specified, but I would imagine that they were made after many rounds of growth
and remodeling. Assembly that started out random evolves into parallel arrays as the
FtsZ protofilaments are inhibited from crossing. Overall I think there are a number of
very interesting observations here, but I am not convinced by the interpretations.

*The model proposed by the reviewer in which FtsZ protofilaments initially assemble on*
*FtsA minirings in a random manner and only form parallel filaments after*
*growth/remodeling, is completely consistent with our new time-lapse results (see Fig.*
*S6). However, the key questions in our opinion are how that assembly evolves and why*
*the FtsZ filaments, once they are organized into the parallel alignments, do not interact*
*laterally to form bundles/pairs when bound to FtsA minirings. Clearly, multiple FtsZ*
*protofilaments have the capacity to interact laterally when attached to membranes, as*

shown e.g. by Milam et al. 2012 and here by us when FtsA* is used instead of FtsA. As
we emphasize in the text and Fig. 7, our model implies that FtsA minirings constrain
FtsZ protofilaments not only so that they remain parallel and do not cross, as the
reviewer suggests, but also so that they remain separated. We believe our
interpretation of the lack of lateral interaction between FtsZ protofilaments is strongly
supported by the EM images and by the in vivo phenotypes. As perhaps we did not
explain things sufficiently clearly, we have modified the text in the Discussion and have
revised Fig. 8, now Fig. 7 (also as suggested by reviewer 2) to make our case clearer
and to pique the interest of readers even further.

Detailed concerns and questions:

Methods for IFM “cells were fixed using one volume of methanol:acetic acid
(4:1) mixture followed by preparation and imaging as described.” The ref “described”
says “as previously described.” Please give the precise conditions used in this study.

*We now describe the conditions in more detail in the Methods and cited a Methods*
*chapter that describes the general protocol in extensive detail.*

The more diffuse width of the Z rings induced by extra FtsA is potentially important but
perhaps not yet convincing. Fig. 2b looks convincing, but S2 less so. The major
difference in S2 seems to be that wt Z rings are overall wider than in ZapA/C. Unless
this can be verified with statistics I suggest removing it. The filamentation phenotype
however is sound.

*This is a good suggestion, particularly as the phenomenon is so variable from cell to*
*cell and Z ring to Z ring. It is also not at all the focus of the paper, which is more on*
*the general cell division inhibition in vivo and the in vitro experiments. As a result, we*
*removed the old Fig. S2, and added statistics about cell lengths instead in another*
*Supplementary Figure (Fig. S3).*

p. 8 l. 8: “antagonizes its assembly into bundles.” I think it is Z rings that are
antagonized. The authors may consider the Z ring a form of bundle, but I think it would
be better to be conservative. Lines 9-10 continue the interpretation that “FtsA inhibits
the assembly of FtsZ into bundles.” It actually inhibits the Z ring as seen by IFM.

*These are good points, and we changed the first instance to say “...antagonizes its*
*assembly into coherent rings”. We also changed the second sentence to be more*
*precise and conservative.*

V208A is presented as another bundling mutant. However I haven’t found any evidence
for its bundling. It antagonized the kil inhibition like L169R, and the authors seem to
have formed the conclusion that this is due to bundling. But they really need to repeat
the EM, pelleting, and GTPase assays that established enhanced bundling for L169R.
E93R was pretty convincing at bundling in Jaiswal, but they might want to repeat this
also.

R174D was originally described to eliminate bundling in Ca, but Moore...Erickson
2016 have recently presented evidence that it bundles just like wt. It is apparently non-
functional and toxic, but it should not be relevant to a bundling hypothesis.

*The reviewer is correct that we don't know if V208A is actually a bundling mutant, and*
*we don't really want to lose the focus of the study by characterizing FtsZ bundling*
*mutants here. As the L169R resistance to FtsA was previously published, and if R174D*
*is indeed capable of bundling, at least in Ca⁺⁺, this leaves only the FtsA-resistance of*
*E93R as new data. As there are plenty of in vitro data, including data we added during*
*the revision, we decided to eliminate Fig. 3 and add the E93R results into Fig. 1, fitting*
*with the theme of ways to resist the toxicity of extra FtsA. The section of text on the*
*mutants was similarly removed, with the E93R results incorporated into the first section*
*of results that includes the ZapA overproduction data, where it fits quite nicely. We*
*kept the Western blot in supplementary (Fig. S2), but removed the mutant proteins from*
*the blot that were not discussed.*

Fig. S5 seems very important. It makes two points. (a) That FtsA must be able to bind
the membrane to be toxic in the DZapA/C background, and (b) that FtsA missing
domain 1C or defective in ATPase are still fully active. It's not really clear how this fits
the bundling hypothesis, but it does seem a potentially important observation.

*The main reason we included these data was to highlight the importance of membrane*
*binding for FtsA's toxic effects in the absence of zapA and zapC, and as a lead-in to*
*doing FtsZ-FtsA experiments on a lipid membrane instead of in solution. We realize,*
*however, that we did not describe Δ1C accurately: the "Δ1C" construct actually is only*
*a small (7 amino acid) deletion within the 1C domain that was previously described in*
*Shiomi et al. 2007, not a deletion of the entire domain as is implied (and the reviewer*
*rightly assumed). This mutant FtsA cannot complement for ftsA, but it is not surprising*
*that it remains toxic, because the deleted residues are in the domain cleft and not near*
*the predicted dimer interface. Based on the reviewer's suggestion, we removed the*
*supplementary figure and instead incorporated the data from the two membrane-*
*binding defective FtsAs (W408E and ΔC15) into Fig. 2a, where we think it fits quite*
*well. We kept the text describing the membrane-binding requirement as a separate*
*section, but focused solely on WT FtsA and these two mutants, as the other mutants are*
*not really relevant to the point we are trying to make.*

The FtsA minirings are pretty spectacular and surprising. Nothing like them has been
seen before. A similar assembly of TmFtsA on lipid monolayers showed relatively
straight pfs, often in pairs. CryoEM of E. coli identified filaments of FtsA running
parallel to FtsZ but closer to the membrane. There was no hint of anything running
perpendicular to FtsZ, but this might be a limitation of the cryoEM in whole E. coli
cytoplasm.

*We agree with the reviewer about the potential limitations of cryo-EM imaging in whole*
*cells and are happy that the reviewer is enthusiastic about the novelty of our findings.*

I doubt that FtsA minirings are physiologically or structurally important in E. coli for a
number of reasons. (a) There are only 1,000 FtsA molecules in the cell. That could
make 83 minirings at most, which could make a continuous row 2,100 nm long, less
than the 3,000 circumference. The Discussion suggested that the minirings might be in
~6 clusters, but then we need something to bridge the clusters to keep the Z ring intact.

*The reviewer brings up the important point that the Z ring cannot consist of extensive*
*arrays of FtsA minirings because there is not enough FtsA in the cell to completely*
*circle the cell. However, several groups have shown by high resolution fluorescence*
*microscopy that the Z ring is patchy and likely discontinuous. That is why, in the*
*Discussion, we proposed the clusters of minirings and their potential sizes. Therefore,*
*there is no need to bridge the proposed clusters of FtsA minirings to keep the Z ring*
*intact, because the Z ring is not continuously uniform around the cell circumference.*
*Moreover, we now know from the preprints of Yang et al. and Bisson-Filho et al. that*
*such clusters may move circumferentially in concert with treadmilling FtsZ filaments,*
*thus covering the entire septal ring area with a mobile set of small protein machines.*
*Consistent with this, even if they formed polymers instead of minirings, 1000 FtsA*
*molecules would at most be able form two continuous polymers around the entire cell*
*circumference. This would be inadequate to anchor a continuous ring of FtsZ polymers*
*and would be far less than the measured ~100 nm thickness of the Z ring. Of course,*
*>1000 ZipA molecules are also available for anchoring FtsZ, but probably they are*
*mostly located within the FtsA-FtsZ patches based on our 2014 3D-SIM paper.*

(b) A major difference between the cell and the lipid monolayer: in the cell there are
only 1,000 FtsA, but in vitro there are an infinite number of soluble FtsA available for
binding. So if the binding affinity is high enough the monolayer in vitro can form a
continuous layer, whereas in vivo the 1,000 A will only be able to cover a small fraction
of the available membrane.

*We thank the reviewer for raising this important point. Naturally, in vitro assays can*
*only approximate an in vivo situation. However, just as there is a larger pool of*
*molecules for binding in our in vitro assay, there is also at least 1000x more surface*
*area of lipids for them to bind than in a cell. In any case, at 0.1 μ M FtsA, we show (and*
*have seen in multiple grids) that minirings are definitely sparser on the monolayer than*
*at 0.5 μ M FtsA, and form in small clusters at 0.1 μ M that may reflect what occurs on*
*the cell membrane. Of course, TEM is not conducive to quantitation, but the trends we*
*observe suggest that the binding of FtsA to the monolayer is limited to some degree by*
*the concentration of the added FtsA. As with the previous point, we believe that it does*
*not argue against the existence of FtsA minirings in cells, only that the clusters are in*
*relatively small patches.*

(c) If the full miniring is not physiologically relevant, it does suggest a more limited
interpretation. Apparently EcFtsZ likes to make curved pfs, and these close to make
minirings when the concentration is sufficiently high on the monolayer.

*If the minirings do not occur in vivo, then we agree that the interpretation would be*
*more limited. But at this point, there is no evidence that they don't occur in vivo. The*
*reviewer is correct that the FtsA minirings resemble the striking FtsZ minirings*
*published 20 years ago by the Erickson group, which inspired us to call the FtsA circles*
*minirings. The low concentrations of FtsA as well as FtsA* seem to assemble into*
*curved protofilaments, like the FtsZ minirings. Nevertheless, there are two key*
*differences between our FtsA minirings and the FtsZ minirings reported in 1996. First,*
*those FtsZ minirings were assembled on a cationic lipid monolayer and normally do not*
*bind to physiological lipids, whereas our FtsA minirings do, through their native*
*amphipathic helices. Second, the FtsZ minirings were separate from each other and*

*seemed to be secondary to (and emerged from) straight FtsZ protofilaments, whereas*
*our FtsA minirings tend to cluster and are the dominant species in many of our*
*conditions. The association of non-bundled FtsZ protofilaments with these FtsA*
*minirings is also entirely consistent with in vivo effects. Therefore, we do not think that*
*the morphological similarity to FtsZ minirings weakens the argument for the potential*
*existence of FtsA minirings in cells. We now make a more explicit reference to FtsZ*
*minirings in the text and note the similarity in their size with FtsA minirings. Note that*
*we did not observe FtsZ minirings under conditions where they could be shown*
*unambiguously (i.e., in the absence of FtsA minirings).*

From the tomogram reconstruction Fig. 6g I calculate a diameter of 23.6 nm for the
mid-point of the ring (based on the 10 nm scale). This gives a spacing of 6.2 nm. This is
much larger than the 4.8 nm spacing in the Sendia xtal structure, and the ~5 nm spacing
of actins and most homologs. Magnifications should be checked and this discrepancy
addressed.

*The reviewer is correct--the scale bar in Fig. 6 was not accurate. Our tomograms were*
*collected at 9,400 × magnification. With calibration, the effective pixel size is 4.5 Å at*
*the specimen level. The diameter of the minirings (from the mid-point of the miniring) is*
*189 Å (42 pixels). Therefore, the spacing between two adjacent subunits is 49.5 Å (189*
*× 3.14 / 12), which is consistent with the ~5 nm spacing between FtsA subunits and the*
*spacing of actin subunits. The figures and methods have been updated and corrected*
*accordingly.*

It is difficult to imagine that the FtsZ filaments could be organized by the circles and
arcs of FtsA. The FtsA-binding peptide of FtsZ is on the end of a flexible linker that can
likely reach 10+ nm in any direction, as well as orient the peptide to match the
orientation any nearby FtsA.

*As mentioned above, this is an important point, but FtsA may induce the compaction or*
*organization of the linker in ways that we do not understand. Another possibility is that*
*the FtsA subunits in each miniring are arranged in such a way that only several have*
*their 2B domains exposed for binding of the FtsA binding peptide of FtsZ. Given that *S.**
*aureus FtsA subunits are twisted relative to each other, then it is possible that *E. coli**
*FtsA subunits are successively rotated in the miniring such that only several subunits*
*can bind FtsZ, potentially imposing spatial limitations on how the FtsZ C-terminal*
*peptide and the FtsZ linker interact with the FtsA minirings; these limitations might be*
*relaxed when FtsA oligomers are no longer constrained to the miniring structures. We*
*have not yet attained sufficient atomic resolution to make that speculation in the*
*manuscript, but it is an intriguing idea. Whatever the structural mechanism, it is clear*
*from our data that the FtsZ filaments are indeed organized by the FtsA minirings, and*
*FtsZ protofilaments bundle significantly in the presence of FtsA* (no minirings); this is*
*the main point of our model. At least in the in vitro system, the flexible linker does not*
*preclude such organization.*

I am not convinced that lipid monolayer structures argue that FtsA inhibits FtsZ
bundling, and that FtsA* is less effective. First, the EM images are anecdotes, and even
then not that convincing. 7f-h shows FtsZ pf pairs, but not the larger bundles.

*We emphasize that Fig. 7f-h (now 6f-h) shows WT FtsA with FtsZ*, not with WT FtsZ as*
*implied by the reviewer, and the default assembly state of FtsZ* is pf pairs. Thus, the*
*almost exclusive formation of pf pairs by FtsZ* (which was true throughout this and*
*other EM grids) is completely expected on monolayers with WT FtsA, and explains the*
*resistance of FtsZ* (and potentially other bundled states of FtsZ, such as those made by*
*E93R) to the normally toxic effects of excess WT FtsA in vivo. On the other hand, when*
*FtsZ* is assembled on lipid monolayers seeded with FtsA*, what are normally pf pairs*
*now associate with each other to become huge bundles/sheets—additional evidence*
*supporting the idea that FtsA* promotes pf bundling.*

Strikingly, one does not see any FtsA minirings in 7f-h. The authors note this lack and
suggest “perhaps because of the FtsZ* polymer mass covering them.” The discussion
ignores the possibility that there may be binding reactions taking place in solution that
modulate the concentrations available for binding the lipid.

*As the reviewer points out, although curved oligomers of what we assume is FtsA are*
*present, few if any FtsA minirings are associated with these FtsZ* doublets at the ratio*
*of 0.5 μ M FtsA to 2.5 μ M FtsZ. This was striking indeed. To explore this further, we*
*examined other grids with FtsA+FtsZ* at a few different concentrations. At 0.5 μ M*
*FtsA and a low concentration (1 μ M) of FtsZ*, many FtsA minirings were present on*
*the lipid monolayer and associated with FtsZ* pfs, although at this low concentration*
*there was a mixture of single and double FtsZ* pfs. Examination of many fields of view*
*indicated that while FtsA minirings often associated with single pfs, they almost never*
*associated with double pfs. We have included a typical image from these new data in*
*Fig. S11a. At 0.5 μ M FtsA + 2.5 μ M FtsZ*, most of the FtsZ* on the monolayer was in*
*the form of double pfs or bundles, yet as we saw before, FtsA arcs and other oligomeric*
*structures were plentiful but very few complete FtsA minirings were present. Further*
*analysis by tomography confirmed this, and we show a typical slice in Fig. S11b.*
*Ideally, we would have tested 1 μ M or higher FtsA with 2.5 μ M FtsZ* to see if the*
*minirings could be restored, but this was not technically possible due to the low*
*concentration of our purified FtsA.*

*We envision two possible scenarios. One, suggested by the reviewer, is that solution*
*interactions between FtsA and FtsZ*(but not WT FtsZ) inhibit FtsA minirings from*
*forming on the lipid monolayer. The general lack of FtsA minirings on lipid monolayers*
*with >2.5 μ M FtsZ* is consistent with this possibility. Another scenario is that FtsZ**
*(and perhaps any bundled FtsZ) may be capable of locally affecting FtsA higher order*
*structure on the lipid monolayer. We favor the latter scenario for 3 reasons. First, in*
*our protocol, FtsA is pre-assembled on the lipid monolayers prior to addition of FtsZ,*
*making it less likely that FtsA-FtsZ solution interactions play a significant role.*
*Second, when we added FtsA and FtsZ simultaneously, where they should be able to*
*interact more extensively in solution, we saw similar patterns on the lipid monolayers*
*(Fig. S7). Third, localized areas of bundled WT FtsZ in the same field as unbundled*
*FtsZ on FtsA minirings are associated with a lack of FtsA minirings, as noted in Fig.*
*4e-g. It may be that bundled FtsZ needs fewer membrane attachment points, or it is*
*more energetically favorable for bundled pfs to bind to the membrane when FtsA is not*
*constrained in the miniring form. Although the relationship between cause and effect is*
*not yet known, and solution effects cannot be ruled out, these experiments demonstrate*
*the value of using mutants of FtsA and FtsZ altered in their higher order assembly*
*states. We are certainly excited to isolate and use such mutants in the future to*

investigate the molecular mechanisms of this potentially important feedback regulation
between FtsA and FtsZ on membranes.

But another important consideration is that the difference in A and A* may be simply
the amount on the monolayer. The binding to the monolayer is probably a cooperative
event, where the weaker A*-A* interaction would mean less bound than the A-A. This
might be approached semi-quantitatively by fluorescent Ab labeling.

*This point at first glance makes logical sense (see our explanation above). However, in*
*addition to the evidence we outlined above, there is also in vivo evidence that FtsA* has*
*at least as high an affinity for the membrane as WT FtsA: (1) FtsA* is as abundant in*
*the membrane vs cytoplasmic fractions of E. coli cells as WT FtsA (Shiomi & Margolin*
*2008); (2) the FtsA* mutation in cis can restore membrane binding to amphipathic*
*helix-defective mutants (Shiomi & Margolin 2008); and (3) FtsA* (and other FtsA*-like*
*mutants) can compete with ZipA better than WT FtsA (Shen & Lutkenhaus 2009;*
*Herricks et al. 2014). Of course, (2) and (3) in particular may be indirect effects, so we*
*tested lipid binding behavior of purified FtsA and FtsA* by quantitating the degree of*
*co-sedimentation with liposomes (now Fig. S10). Although a considerable fraction of*
*purified FtsA/FtsA* sedimented in the absence of liposomes, the addition of liposomes*
*caused 100% of either protein to reproducibly cosediment, indicating a robust*
*interaction. In contrast, there was no increase in pelleting of an MTS-defective FtsA*
*(FtsAΔC15) with liposomes. Together, these results strongly support the idea that*
*FtsA* is not deficient in lipid binding ability compared with WT FtsA. In addition, we*
*showed that protofilaments of WT FtsZ bundle extensively on lipid monolayers with*
*FtsA* relative to FtsA. Although the molecular mechanism is not yet known, it is hard*
*to imagine how markedly increased interactions between FtsZ protofilaments at the*
*membrane could occur if their membrane anchors were less concentrated.*

At the very least this idea should be pursued by increasing the concentration of A*. It
might even form complete minirings if its surface concentration were raised to match
that of A.

*We agree with the reviewer that increasing FtsA* concentration would tend to favor*
*more complete minirings, but not necessarily because of increased affinity for the*
*membrane. Indeed an increase in minirings over arcs or smaller oligomers could also*
*arise from increasing A*-A* subunit interactions, which would help to elongate arcs*
*and close them, making more minirings. This might occur via a mechanism similar to*
*how increased concentration of FtsZ favors formation of FtsZ minirings on cationic*
*lipid monolayers. Therefore, an increased frequency of closed FtsA* minirings at*
*higher protein concentrations would not distinguish between more membrane binding*
*by FtsA* or simply more FtsA*-FtsA* interactions independent of membrane binding.*
*It would also be technically challenging for us to increase FtsA* concentrations, as a*
*much higher concentration of active purified FtsA* (or FtsA) has not been attainable.*

A related question is the order of addition. Fig. 7e caption says + “0.5 μM A* + 5 μM
Z*” while p.14 1.8-9 says “we added 0.5 μM A* to the monolayer and then added 5 μM
Z*”. This may make a big difference, whether the A and Z are first equilibrated in
solution, or A* is first equilibrated with the monolayer.

*We apologize for the ambiguity in the description to what is now Fig. 6. The “+”*
*means that FtsZ* was added to the monolayer after FtsA*, as described in the Methods.*
*This was done because we wanted to tether FtsA(FtsA*) to the membrane first so we*
*could confirm their oligomerization patterns before adding FtsZ/FtsZ*. However, the*
*reviewer makes an excellent point about modifying the order of addition, which was*
*suggested also by reviewer 2. In new experiments, we added FtsA and FtsZ*
*simultaneously to the lipid monolayers and found that the FtsA minirings and FtsZ*
*protofilaments were organized into patterns similar to those from lipid monolayers with*
*pre-assembled FtsA (Fig. S7). This indicates that the order of addition may not matter*
*that much for the organization of these structures.*

Reviewer #2 (Remarks to the Author):

The paper provides new insights into the oligomerization of the actin homolog FtsA
protein and its implication for the bundling of FtsZ filaments. It combines in vivo data
on the cell viability and FtsZ ring morphology at different genetic backgrounds with the
biochemical reconstitution of the purified proteins on a lipid monolayer. The later
allows for a high resolution characterization of FtsA oligomeric state by EM. Together
the authors suggest a model by which a higher-ordered state of FtsA (membrane-bound
minirings) antagonizes bundling of FtsZ filaments.

The manuscript is organized in two parts: the first part is dedicated to in vivo
experiments where the phenotypes and viability of the strains overexpressing FtsA, FtsZ
and Zap(s) proteins are characterized. These experiments are based on previous
observations that although overexpression of FtsA is lethal, its toxicity can be
suppressed by either co-overexpression of FtsZ [Dai & Lutkenhaus 1992, Dewar et al
1992] or by the presence of hyperbundled FtsZ (FtsZ*) [Haeusser et al 2015]. The
authors substantiate these findings by showing that the negative influence of FtsA on
FtsZ bundles is counterbalanced by the bundling activity of ZapA and ZapC.
Accordingly, the toxic effect of overexpressed ZapA can be compensated for by co-
overexpression of FtsA.

Overall, the results shown in this part are convincing, however it is disappointing that
the paper does not include more recent imaging techniques or image analysis to better
quantify the influence of ZapA, ZapC and FtsA on FtsZ filament bundling. Apart from
the not-convincing model (see below), I think this is biggest weakness of the paper. I
think it would great benefit from one of the three following approaches:

Super-resolution microscopy of FtsZ filaments in wt and mutant cells. This is pretty
much standard these days and as ZapA, ZapC and modest increase of intracellular FtsA
seem to have only subtle effects on FtsZ ring architecture (see Fig. S2), their influence
can be visualized much more convincingly at higher spatial resolution.

live-cell imaging. It is not clear to me why this is not standard nowadays. A number of
recent high-impact papers have demonstrated very beautifully that by just look at the
dynamics of proteins in vivo a much better understanding of intracellular organization
can be achieved (see for example Garner et al, Domínguez-Escobar et al, van Teeffelen
et al, 2011 or Bisson Filho et al, Yang et al, 2016 (bioRxiv). This will also be true for
this paper.

better statistics/image analysis. Fig. S2 is very disappointing, do these lines represent
fluorescence intensity profiles along individual cells? The intensity distributions of FtsZ
rings greatly vary between different cells and during the cell cycle. Accordingly, such a
representation is rather meaningless. The authors should normalize the intensity profile
for different cell lengths, average the intensity profile over many cells and normalize the
intensity profiles. The authors might find this tool useful:

[https://sils.fnwi.uva.nl/bcb/objectj/examples/Coli-Inspector/Coli-Inspector-MD/coli-](https://sils.fnwi.uva.nl/bcb/objectj/examples/Coli-Inspector/Coli-Inspector-MD/coli-inspector.html)
[inspector.html](https://sils.fnwi.uva.nl/bcb/objectj/examples/Coli-Inspector/Coli-Inspector-MD/coli-inspector.html)

*We are happy that the reviewer thought that the in vivo results were convincing.*
*However, it is not clear how doing higher resolution studies would strengthen the*
*conclusions significantly; any changes in Z ring thickness, for example, might suggest,*
*but not really prove, that bundling had changed. Moreover, ring thickness in zapA*
*mutants vs WT cells has already been reported (Dajkovic et al., 2010). However, we*
*agree with reviewer 1 that the micrographs showing filamentation (along with the spot*
*dilutions) are sufficient to suggest that FtsA antagonizes FtsZ bundling in vivo. We also*
*agree with this reviewer (and reviewer 1) that Fig. S2 was weak because Z ring widths*
*and intensities are so highly variable among and within the filamentous cells.*
*Therefore, we have removed Fig. S2 and replaced it with a graphical representation of*
*cell lengths (Fig. S3).*

In the second part, perform biochemical experiments and electron microscopy to
elucidate a potential molecular mechanism by which FtsA prevents FtsZ bundling.
By using purified proteins and lipid monolayers, the authors found that FtsA is able to
organize into minirings on a membrane. This is first visualization of membrane-bound
filaments of E. coli FtsA and the formation of FtsA minirings is in contrast to previous
observations of T. maritima FtsA forming long straight filaments [Szwedziak et al
2012]. When polymerized FtsZ was added, FtsZ filaments were found to be recruited to
the FtsA covered membrane, forming long, non-bundled filaments. This data is
beautiful and will certainly attract the attention of the field.

*We are glad that the reviewer appreciates the quality and importance of our findings.*

However, the second main problem of the paper is that the authors fail to convincingly
interpret their observations and to propose a unifying model. Especially in light of
recent findings of dynamic treadmilling of FtsZ and FtsA in vivo [Bisson Filho et al,
Yang et al, 2016 (bioRxiv)] and in vitro [Loose & Mitchison 2014], the authors need to
do a better job in putting their observations in perspective.

*We thank the reviewer for making us aware of the two preprints on treadmilling in*
*BioRxiv. We have added a sentence in the introduction about FtsZ treadmilling in vitro*
*and in vivo and have cited those preprints as well as Loose & Mitchison. We have*
*included FtsZ treadmilling in the model (Fig. 7) as well. We also briefly mention FtsZ*
*treadmilling again in the Discussion and how FtsA minirings might be involved,*
*although at this point a more detailed discussion or a unifying model would be much*
*too speculative.*

For example, the authors appear to suggest that FtsA disrupts FtsZ bundles by
assembling into minirings, which keep FtsZ filaments at a certain distance. This

fragmentation of FtsZ bundles is important to perform cell division. However, as
suggested in the last figure, at in vivo conditions, FtsA might not form rings but arcs or
shorter oligomers. Accordingly, it must disrupt FtsZ bundles using a different
mechanism than via FtsA minirings. Instead these minirings might represent the toxic
FtsA species able to prevent cell division.

*We think that our original model figure gave the wrong impression to the reviewer*
*about the form of FtsA that disrupts FtsZ bundles. In Fig. 8 (now Fig. 7), we proposed*
*that FtsA minirings form at the initial setup stage of the Z ring, anchoring FtsZ*
*protofilaments to the membrane but preventing their lateral interactions. Based on our*
*EM data and previous genetic data, we proposed that these minirings represent the*
*“locked” (and therefore toxic) form of FtsA and they do not allow for progression of*
*cell division until they are “unlocked” by some factor that induces breakup of the*
*minirings into arcs or shorter oligomers, which in turn relieves the antagonism of FtsZ*
*bundling initially imposed by the minirings. So the reviewer is correct in that FtsA*
*likely is not always in a miniring form.*

*Importantly, the reviewer states that the breakup of the minirings “must disrupt*
*FtsZ bundles using a different mechanism than via FtsA minirings”. However, it is*
*crucial to emphasize that we proposed the opposite: FtsA miniring disruption promotes*
*FtsZ bundling (by relieving the antagonism of FtsZ bundling imposed by FtsA minirings*
*and allowing lateral interactions between FtsZ protofilaments to occur). This model is*
*supported by our lipid monolayer assays with FtsA*, which is known to accelerate*
*septum formation in vivo and seems to be constitutively disrupted for miniring*
*formation in our assays. We speculate that this acceleration of septation potentially*
*bypasses the initial Z-ring setup phase normally organized by the “locked” FtsA*
*minirings. When at the same concentrations as WT FtsA, FtsA* on lipid monolayers*
*forms mainly arcs and smaller oligomers instead of minirings, allowing FtsZ to form*
*bundles. Although we did make these points about releasing constraints by FtsA*
*minirings in our submitted manuscript (and were interpreted correctly by reviewers 1*
*and 3), we nevertheless have significantly modified the model figure and text to make*
*the points more clear. We do agree with the reviewer that FtsA minirings are likely an*
*important reason why FtsA overproduction is toxic; this idea was/is also discussed in*
*the manuscript, and fits the notion that preventing FtsZ bundling can ultimately block*
*septation at the early setup stage (see new Fig. 7). The importance of FtsZ filament*
*condensation for septum formation was also previously reported by Lan et al. 2009.*

In summary, this paper further substantiates previous observations that FtsA has the
ability to disrupt FtsZ filaments or FtsZ filament bundling. The EM data sheds light on
E. coli FtsA structures. However, the paper fails to propose a convincing model that
could help understand how FtsA, FtsZ and Zaps interact to coordinate cell division. For
publication, major revisions are needed.

*Our EM data not only shed light on the structures that FtsA forms on a lipid membrane,*
*but also allow us to propose, for the first time, how these higher order structures of*
*FtsA can control the higher order state of assembled FtsZ. We did not include Zap*
*proteins in our model because it is not known precisely how Zaps contribute to FtsZ*
*assembly state in the context of the membrane (see also reviewer 1’s comments about*
*the potential diversity of Zap-mediated FtsZ bundling). We also did not include ZipA,*
*which as an essential protein is more important than Zaps, for similar reasons. As a*
*result, to avoid unnecessary speculation and to keep the focus on what we studied in*

*vitro, namely FtsZ and FtsA, we keep the model focused on these proteins, but have*
*modified it to make it more detailed. For example, we added a diagram of what FtsZ,*
*FtsA and Zaps/ZipA might look like in cells at two different stages of FtsA oligomeric*
*states (Fig. 7d), and kept the sentence mentioning how Zaps could integrate into the*
*coordination of cell division in the Discussion.*

In addition, I do have a number of other questions that I hope the authors can address:

1st part, in vivo experiments

page 7, line 5: the picture of the gel appears to have been stitched together from
multiple experiments. The authors should either make it obvious that it originates from
different gels or repeat the experiment with all samples on one gel.

*We have now demarcated the gel lanes in Fig. S1 that originated from separate gels.*

page 7, line 13: Evidence in lines 13-22 is very weak. Fig. S2: Do the lines correspond
to three individual cells? FtsZ ring widths should be plotted as a function of the cell
length and averaged over many cells to make a strong statement. For example, if Zaps
are bundling proteins it does not make much sense that the rings appear to be tighter in
those cells lacking Zaps.

*This figure was removed, and replaced with a plot of cell lengths in the different*
*mutants/conditions (Fig. S3).*

page 9, line 17: Does FtsA prevent FtsZ polymerization and bundling in solution or on a
membrane? While the present manuscript suggests that FtsA would only be active when
both proteins are in contact with each other on the membrane, in a previous paper
[Beuria et al 2009], the lab of Margolin showed that FtsA shortens FtsZ filaments in
solution.

*The Beuria et al. paper described the effects of FtsA*, not FtsA, and given that we only*
*saw FtsZ but never saw any credible FtsA/FtsA* structures on EM grids from solution,*
*we are not sure how the effect is mediated in solution. In addition, in that study, WT*
*FtsA had no effect on FtsZ in solution. It could be that the overall reduction in*
*sedimentable FtsZ polymer mass by FtsA* reported in Beuria et al. was some type of*
*inhibition of FtsZ protofilament bundling in solution, which then led to protofilament*
*breakage or instability. It is also possible that membrane-bound FtsA* or FtsA has an*
*FtsZ depolymerizing activity that is in equilibrium with FtsZ filament growth and*
*remodeling (via treadmilling), resulting in no net FtsZ disassembly as long as GTP is*
*present. Basically, in the absence of membranes, without the membrane targeting*
*sequence engaged in binding lipids, FtsA is likely to have very different properties than*
*when it is in solution.*

page 9, line 24: “FtsA Δ 1C, lacking the 1C domain important for self-interaction and
recruitment of late division proteins, ..., inhibited growth when induced, like WT
FtsA”. This suggests that FtsA does not need to form minirings to exert its antagonistic
effects on FtsZ polymer bundling.

*As mentioned in our response to reviewer 1, this sentence was an error on our part: this*
*particular construct is deleted for only 7 amino acids within IC (from V127 to Q133,*
*cited in Shiomi & Margolin, 2007), so the domain is not lacking, just defective. This*
*construct remains toxic (like WT FtsA) probably because the dimer interface is still*
*intact: the 7 residues deleted are in the cleft of the IC domain, which might be*
*important for interacting with later division proteins, but are not at the dimer interface.*
*Because the main point of this experiment was to show whether or not membrane*
*binding was needed for FtsA's toxicity in the $\Delta zapA \Delta zapC$ mutant, we deleted this*
*figure but moved the data comparing the two membrane-binding mutants from the*
*supplementary figure to Fig. 2a. We plan to investigate the assembly properties of many*
*interesting FtsA mutants, including those in the IC domain, in future studies.*

2nd part, in vitro EM

For their biochemical experiments, the authors use N-terminal 6xHis-tagged FtsA
expressed from pET28, which also contains a thrombin site [Geissler et al 2003] It is
well known that the Polyhistidine can alter the solubility and assembly properties of
protein polymers [Petek & Mullins 2014], therefore it is important that the authors
demonstrate that FtsA still forms membrane-bound minirings upon removal of the
6xHis-tag.

*Until 2014, the biochemical characterization of E. coli FtsA had been hampered by its*
*tendency to aggregate after purification. Now the purification protocols have been*
*optimized and we are able to purify the His-tagged WT FtsA at the concentrations (~4*
*μM) that allow performing basic biochemical assays (Herricks et al., 2014). To address*
*the reviewer's concern, we cleaved off the His₆ tag (using thrombin) from two protein*
*preparations: (a) purified 3.5 μM His₆-FtsA, and (b) His₆-FtsA(Y139D), a gain of*
*function mutant (Herricks et al., 2014) that we can purify at significantly higher*
*concentrations (up to 12 μM). This mutant protein mostly forms minirings and arcs on*
*lipid monolayers that are similar to those formed by WT FtsA (unpublished data). We*
*found by anti-His₆ immunoblotting that after thrombin treatment, most of either cleaved*
*protein lacked a His₆ tag, although a considerable amount of protein degradation*
*products were present in both preparations. When we added His₆-tag-cleaved WT FtsA*
*to lipid monolayers, we saw only a few minirings and a large amount of background*
*material, presumably due to the presence of a large proportion of degradation products*
*that could have interfered with the assay. However the Y139D mutant, which initially*
*was 3x more concentrated, still formed abundant mini-*
*rings (see accompanying TEM image) that were identical*
*in morphology to those formed by His₆-FtsA (or His₆-*
*FtsAY139D). Therefore, we see no evidence that*
*assembly of FtsA into minirings is promoted or otherwise*
*affected by the presence of the His₆-tag. This is*
*consistent with the marked deficiency in the ability of*
*His₆-tagged FtsA* to form minirings, consistent with two-*
*hybrid assay phenotypes. Moreover, our His-tagged WT*
*FtsA, when mixed with FtsZ, often strongly resembles*
*the swirl patterns observed on membranes by Loose and*
*Mitchison (2014), e.g. Fig. 4h, Fig. S6b. Finally, the S.*
*aureus FtsA used for determining the atomic structure*
*had an intact N-terminal His₆ tag similar to ours. We*

thank the reviewer for pointing us to the Petek & Mullins paper, which mentions that
polyhistidine tags can alter assembly properties of actin-like polymers; unfortunately,
the statement cites unpublished work.

Expression of FtsA from pDSW210F-ftsA at 500 μ M IPTG is toxic. How could the
authors purify FtsA from pET28 at 1 mM IPTG?

*Expression of FtsA from pDSW210F-ftsA at high IPTG concentrations is indeed toxic,*
*because even though cells continue to grow, they no longer divide and so produce*
*filaments that are viable for several generations. Similarly, with pET28a-FtsA in the*
*C43 strain, the cells start to filament after IPTG induction, but by the time most*
*filaments would die, we have already harvested the cells. (We also added in this section*
*that for FtsA overproduction we used C43, which is a derivative of BL21(DE3), not*
*BL21(DE3) itself).*

Why their FtsZ filament are so long, more than 1 μ m in length? This is much longer than
observed previously. If FtsA is increasing FtsZ dynamics and decreasing FtsZ stability
[Beuria et al 2009, Loose & Mitchison 2014], should the filaments not be much shorter
than that? Would shorter filaments bind to FtsA with a random orientation?

*The reviewer brings up a good point. The length of the filaments is in part due to the*
*long incubation time in the presence of GTP. As mentioned in the response to reviewer*
*1, in new time-lapse experiments we found that at shorter (<1 min) incubation times*
*with FtsZ, the FtsZ filaments are shorter, on the order of 100-300 nm in length, sparser,*
*and less aligned (Fig. S6). By 5 min after addition of FtsZ to the pre-assembled FtsA*
*minirings on the lipid monolayer, protofilaments were more concentrated and already*
*aligned in parallel, similar to their appearance after the longer incubation periods. This*
*indicates that FtsZ filament reorganization on FtsA minirings occurs rapidly and*
*evolves toward filament alignment. Loose and Mitchison also observed long filaments*
*and large micron-scale circular swirls of FtsZ and FtsA that are consistent this idea*
*and our images. One likely possibility is that factors in vivo can sever or otherwise*
*limit the length of FtsZ polymers on FtsA/membranes, and these factors are not present*
*in the purified system.*

The density of FtsA on the membrane depends on the absolute amount of protein
present in solution i.e. the surface to volume ratio at a given concentration. What was
the buffer volume and the surface area in the experiment?

*The buffer volume in the Teflon block well is 90 μ l or 90 mm^3 ; the surface area of the 3*
*mm diameter EM grid, and thus the lipid monolayer, is 7.2 mm^2 . So the surface to*
*volume ratio is ~ 12 . We added the well volume to the Methods.*

Is FtsA oligomerization important for membrane binding? If yes, one should see a
highly sigmoidal curve in FtsA-titration experiments (amount of membrane-bound FtsA
vs. FtsA concentration).

*We have not done the suggested titration experiment, but we do know from both*
*liposome sedimentation (see response to reviewer 1 and new Fig. S10) and the high*
*density of FtsA* on the lipid monolayers as seen in many EM fields that FtsA* seems to*
*bind to membranes as well as WT FtsA, suggesting that significant oligomerization of E.*

*E. coli FtsA may not be required for efficient membrane binding. In vivo, FtsA**
*fractionated to cell membranes as well as or better than WT FtsA, and the cell division*
*function of the membrane binding-defective W408E mutant can be largely restored by*
*introducing the R286W (FtsA*) mutation in cis, suggesting that less oligomerization*
*may even enhance membrane binding. So at this point, there is no compelling evidence*
*that E. coli FtsA needs to oligomerize extensively in order to bind the membrane.*

On the EM micrographs it looks as if there was much more FtsA bound to the
membrane than FtsZ. Given that they authors say they used physiological protein
concentration ratios, could the authors comment on that apparent discrepancy?

*We surmise that the apparent abundance of FtsA molecules on the lipid monolayer*
*relative to FtsZ, despite the higher amount of FtsZ in solution relative to FtsA (as it is in*
*E. coli cells), is likely due to the strong affinity of FtsA for the membrane compared to*
*the affinity of FtsZ for FtsA. We know that some polymerized FtsZ remains in solution*
*after incubation with the lipid monolayers.*

The authors used a rather long incubation times for FtsA and ATP (1 hour total),
especially compared to Loose & Mitchison 2016, where the dynamic behavior of the
proteins were found within at earlier time points. Is the formation of FtsA time-
dependent? Does it depend on the order by which the proteins were added?

*To address the question of timing, we pre-assembled FtsA on the lipid monolayers as*
*before, but imaged samples at very short times after adding FtsZ and initiating its*
*assembly with GTP. We found that at the 30-60 second time points, FtsA rings were*
*present and FtsZ polymers were relatively short and not yet all aligned. By the 5-*
*minute time point, the FtsZ polymers had reorganized into the aligned but unbundled*
*state as we have already shown for the longer incubation times. We can conclude that*
*the organization and alignment occurs rapidly, on a time scale consistent with what*
*occurs in vivo. Typical images from a time-lapse experiment are shown in Fig. S6.*

*We also added FtsA and FtsZ simultaneously, and saw patterns (FtsA minirings + FtsZ*
*protofilaments) that were similar to those under our standard FtsA first, then FtsZ*
*protocol. This was added to the text and to the supplementary figures (Fig. S7), and*
*indicates that preassembly of FtsA before addition of FtsZ is not required for FtsA to*
*form minirings.*

How does the diameter of the ring, and therefore the distance between two FtsZ binding
sites, compare to the distance between the corresponding FtsZ monomers?

*The distance between two FtsZ binding sites on the FtsA miniring is ~20 μm. The 4 nm*
*spacing between FtsZ subunits would suggest that 4-5 subunits of FtsZ lie between the*
*two FtsA-bound FtsZ subunits.*

Discussion

Fig. 8. can be improved. For example the authors could include an illustration of how
FtsA minirings might be organized in the cell (see page 15, line 22-page 16, line 2).

*In response to this suggestion, we added another part to the model figure (now Fig. 7d)*
*that illustrates how FtsA and FtsZ (as well as ZipA/Zaps) might look in two*

*representative patches on the cell membrane at initial setup phase of the septal ring and*
*after the putative “unlocking” step.*

Reviewer #3 (Remarks to the Author):

In this paper, the authors characterize the role of FtsA and FtsZ in E. coli divisome
assembly. The authors present genetic evidence using a number of mutants that promote
or inhibit FtsZ bundling to support the hypothesis that FtsA antagonizes FtsZ bundling
in living cells. Remarkably, in vitro structural studies using electron microscopy of
purified components show that physiologic concentrations of FtsA assemble into
minirings on lipid monolayers and that these rings guide the assembly and organization
of FtsZ protofilaments and prevent their bundling. The authors then focus on a
hypermorphic mutant of FtsA that accelerates septum formation and bypasses the
requirement of ZipA. In vitro results show that bound FtsA* does not form minirings
but assembles into small oligomers that promote assembly of disorganized and bundled
FtsZ protofilaments. Taken together, the in vivo and in vitro evidence supports the
hypothesis that FtsA acts to antagonize FtsZ bundling and that the
oligomeric state of FtsA is important to influence the organization FtsZ. The authors
present a working model where the oligomeric state of FtsA serves as a molecular lock
in early stages of protoring formation to keep FtsZ unbundled until proteins are
recruited during maturation of the divisome. FtsA is then converted to non-ring forms
(presumably by other divisome proteins) which then subsequently releases constraints
on FtsZ orientation and bundling.

This is an important and interesting paper that uses a comprehensive approach with
genetic, biochemical and structural methods to establish the roles of FtsA and FtsZ in
bacterial cell division machinery. Admittedly, this mechanism may not be universal to
all bacteria. The results complement and extend previous light microscope observations
by Loose and Mitchison on the importance of FtsA in tethering FtsZ to lipid
monolayers. The paper is well written and clearly lays a path of important genetic and
structural results to support their hypothesis that FtsA performs a more regulatory role
and acts to antagonize the bundling of FtsZ protofilaments. However, there are a
number of major and minor points that need to be addressed prior to publication.

*We thank the reviewer for appreciating the significance and quality of our work.*

Major:

The authors present beautiful tomography and 3D volume averaging of FtsA and FtsZ
organization in vitro. The resulting tomograms clearly show minirings of FtsA bound to
lipid monolayers. Each FtsA miniring is formed from twelve subunits. Unbundled FtsZ
protofilaments bind to two of the twelve subunits of the membrane-distal region of the
FtsA ring. However, the 2D images of the negative stained samples of FtsA* bound to
lipid bilayers were less convincing. The study would benefit from tomography of FtsA*
with FtsZ to visualize the underlying FtsA* fragments more clearly and to understand
the interaction of FtsZ with these short oligomers. For example, does FtsZ still maintain
binding to two FtsA* subunits? This would bring more detailed and realistic
information to support the model presented in Figure 8b.

*We agree with the reviewer that a higher resolution structure of an FtsZ protofilament*
*pair/bundle with FtsA* oligomers would help clarify the second part of the model. We*
*analyzed tomographic slices of lipid monolayers with FtsA*+FtsZ and observed many*
*short oligomers of FtsA* in the membrane proximal plane and extensive bundling of*
*FtsZ above. We added a supplemental figure (Fig. S9) containing representative*
*images. However, we were unable to obtain more detailed structural information,*
*because it was not possible to perform averaging with such heterogeneous structures.*
*We favor the idea that FtsA oligomers, being less organized into minirings than WT*
*FtsA, are more densely packed on the lipid monolayer than WT FtsA; this higher*
*density of membrane tethers enhances the ability of FtsZ protofilaments to interact*
*laterally. These EM images inspired us to propose how this might work in greater*
*detail in the model figure (Fig. 7c-d).*

Use of negative stain samples for high resolution averaging and docking of crystal
structure is not typical due to the fact that these samples are dehydrated and heavily
stained. The authors correctly point this fact out on page 12 (lines 19-1) where they
point out that higher resolution structures would be needed to confirm the model. The
limitations using negative stain samples is also eluded to in the discussion on page 16
(lines 19-21) where the authors point out that the staining method could lead to
compression of flexible linkers in the protein. The use of negative stain samples in this
study was appropriate in that it provided a method to illustrate the overall organization
of FtsA and FtsZ under various conditions. A statement in the discussion that future
studies using cryoEM of vitrified samples will be important to obtain higher resolution
structures to overcome the limitations of the negative stain technique.

*We agree with the reviewer on this important point and have added the suggested*
*statement to the discussion.*

The organization of FtsA minirings in vitro presented in this paper was remarkable, but
as the authors point out there is no direct evidence that minirings are present in cells.
The discussion could be developed further by proposing that future studies using
cryoET may be useful for visualizing the cell division machinery in vivo, similar to
what was done with *Caulobacter crescentus* (Li et al., 2007). Although *E. coli* is too
large for whole mount cryoET, methods such as vitreous cryosectioning (CEMOVIS) or
cryo FIBSEM may offer important avenues for identifying these structures in a near-
native state.

*We agree with the reviewer on these points and have added a reference to future work*
*using these methods at the end of the Discussion.*

Minor:

Discussion: page 16, line 7: FtsA should be FtsA*

*Thanks for catching this error—corrected.*

Figure 4. I would expect Figures 4c and 4f to look more similar. To my eye Figure 4f
looks more like Figure 4d. Some comment about variability of the preparation should be
mentioned.

*This is a good point, and there was, not surprisingly, variability between experiments*
*and among different fields in the EM. This is now mentioned in the text.*

Figure 5. Figure 5d. The panel shows 0.5um FtsA + 5uM FtsZ on a grid without a lipid
monolayer. The text says ‘only FtsZ filaments were detected’. Some statement about the
absence of FtsA rings should be added. There is a lot of background material that could
be unassembled FtsA and there does appear to be more filaments than in the control
(Figure 5a) were FtsZ is added to lipid bilayer alone.

*The reviewer is correct about the general lack of FtsA rings in the absence of the lipid*
*monolayer, and we added this to the sentence. We do sometimes observe irregular*
*“rings” of diverse sizes that may comprise FtsA that remains tightly bound to lipid*
*membranes from the original protein isolation. However, the rarity of these structures*
*makes it difficult to confirm that they are actually FtsA, so we do not mention them.*

*The higher number of FtsZ filaments in (d) is likely because the grid for (d) was glow*
*discharged, while the grid for (a) has the lipid monolayer. Glow discharging the grid*
*allows efficient binding of soluble FtsZ filaments.*

Movie S2 was mentioned in the Methods section but not mentioned in the Results
section of the text. Perhaps it could go on page 12 line 19: (Figure 6h; movie S2). The
movie could also be mentioned in the legend of Figure 6h.

*We thank the reviewer for this suggestion and now have cited movie S2 in the two*
*places suggested.*

Figure 7. It would be useful to have arrows pointing to arcs and short curved
protofilaments or even an inset at higher mag in figure 7b. It was difficult to see these
structures at the magnification of the image.

*We thank the reviewer for this suggestion. An inset at 2.5x magnification was added to*
*Fig. 6b (old 7b), making the arcs much easier to see.*

*We have done tomography on some of these grids and it is clear that the FtsA*
*underneath the protofilaments of FtsZ* at this concentration is oligomeric with few*
*minirings present. This suggests that FtsZ* resists FtsA toxicity in vivo because FtsZ**
*somehow antagonizes assembly of FtsA minirings. The correlation between bundled*
*FtsZ and the lack of FtsA minirings is also evident in certain areas of the monolayer in*
*the experiments with FtsA+FtsZ (Fig. 4e-g) and in experiments with FtsA* + FtsZ,*
*where the miniring-free FtsA* associates with bundled FtsZ (see Fig. S9). Although the*
*molecular mechanism for this correlation is not yet known, the evidence suggests a*
*positive feedback loop in which disassembly of FtsA minirings on the membrane*
*enhances lateral interactions between FtsZ protofilaments, which in turn spur further*
*disassembly of FtsA oligomers (see also response to reviewer 2). Such a feedback loop*
*in vivo would tend to drive the septation pathway irreversibly forward. However, we*
*feel that this interesting idea is speculative at this point without further investigation,*
*particularly using genetics, so we did not give it too much emphasis in the text.*

Figures S6 and S7 should have scale bars on the images. FigS6 compares periodicity
from a tomographic slice in the top panel (a) with 2D projection of the mutant in (c).
*We added scale bars to both Figures (now S5 and S8). We did not do tomography with*
*the FtsA*-FtsZ* sample (2D projection shown in panel c) because the nature of the*
*large bundled structures was already clearly evident in the projections.*
Technical details in the methods on Page 21 lines 13-22. “Tomography data collection
and reconstruction”
-The pixel size at which the data were acquired should be explicitly mentioned. This is
important for the reader to know the resolution of the tomograms and subvolume
averages. Since SerialEM was used to acquire the data, this information will be in the
image header.
*There was a small error in the size calibration (see response to reviewer 1). This was*
*corrected, and a sentence specifying the pixel size (4.5 Å) was added to the Methods.*
-It is unusual to collect images of negative stained samples with such a large defocus (-
6µm). This is typically used for unstained samples in cryoEM. Was CTF correction used
on these data? The authors incorrectly cite a program “TOMOAUTO” as Kremer et al.
and Xiong et al. The Kremer et al. paper references the IMOD software package for
alignment and reconstruction and the Xiong et al. references methods for CTF
correction.
*The reviewer is correct. The defocus is relatively high for a negative stained sample.*
*However, in this particular study, we used higher defocus to increase contrast for a*
*better visualization of the minirings and filaments. We did not use CTF correction in*
*IMOD. In fact, we did not use IMOD for alignment and reconstruction; instead, we*
*used TOMOAUTO only for assembly of drift-corrected stacks. We then used*
*tomographic package Protomo (Winkler & Taylor, 2006) for marker free alignment and*
*weighted back projection. The correct references are now cited.*
-Similarly, on page 21 line 21 there is a (5) at the end of the sentence about alignment
and reconstruction. Perhaps this is where the Kremer and Xiong references should be
correctly cited.
*We thank the reviewer for noticing this formatting error. We have cited the correct*
*Winkler and Taylor reference here.*
Page 17, line 23 should read “both of which”
*Corrected.*

REVIEWERS' COMMENTS:

Reviewer #1 (Remarks to the Author):

The authors have submitted a 21-page rebuttal to the concerns raised in the original reviews. Some new data (not very much) and clarifications have improved the mss. Much of the rebuttal was just that, responses that often invite further discussion. But 21 pages is enough. Since the journal's policy is to publish the full reviews and response along with the paper, I would think it satisfactory now to suggest that many of the concerns and alternative interpretations raised by myself and the other referees remain valid. Readers can read these original concerns and the rebuttal and decide for themselves how much weight to give them.

Reviewer #2 (Remarks to the Author):

Overall, the authors have addressed my concerns and have improved the manuscript. I think their model is not completely convincing, but it contributes to the currently lively discussion about the mechanism of bacterial cell division.

Reviewer #3 (Remarks to the Author):

Escherichia coli FtsA assembles into polymeric rings on membranes that align FtsZ protofilaments and antagonize their lateral interactions. Krupka et al.

In this revised manuscript, Krupka et al. address all of the questions and concerns of the reviewers and provide additional experiments that make important contributions to their working model of the role of FtsA and FtsZ in *E. coli* divisome assembly. The manuscript is greatly improved and will certainly attract attention to a wide readership.

In particular, the authors presented beautiful tomography data of FtsA and FtsZ organization. The resulting tomograms clearly show minirings of FtsA bound to lipid monolayers and unbundled FtsZ protofilaments bound to the membrane-distal region of the FtsA ring. However the 2D images of FtsA* + FtsZ were less convincing to myself and the other reviewers. The authors have now provided new tomographic data in a new supplemental figure (S9) that I feel clearly shows the short FtsA* oligomers proximal to the substrate and bundled FtsZ above. I completely agree that more detailed structural information using image averaging would not be possible with such heterogeneous structures. The organizational differences between the data in figure 5 (FtsZ protofilaments on top of FtsA minirings) and the new S9 (FtsA* + FtsZ) is remarkable and strengthens their working model in the new Figure 7.

The authors also addressed and corrected all of the minor points and errors in my review. The addition of higher magnification insets (Figs 4 and 6) and arrows help to identify the structures of interest more clearly, especially to those unfamiliar to negative stain EM.

The text has also been expanded to address the issues raised about using negative stain samples for this type of high resolution work. There are limitations to the use of negative stain that the authors have now added a statement proposing the importance of future studies using cryoEM to overcome these limitations.

Although organization of FtsA minirings presented in the paper was remarkable, there is no direct evidence that minirings are actually present in cells. The authors have expanded the discussion to

include a statement to future work using cryoET methods to provide important avenues for identifying these structures in a near-native state.

Overall, I feel the authors addressed all of my questions and concerns and provide a thorough and comprehensive revision that will make important contributions to the bacterial divisome field but will also be of interest to a wide readership.

Response to reviewers' comments (our responses are in italics):

Reviewer #1:

The authors have submitted a 21-page rebuttal to the concerns raised in the original reviews. Some new data (not very much) and clarifications have improved the mss. Much of the rebuttal was just that, responses that often invite further discussion. But 21 pages is enough. Since the journal's policy is to publish the full reviews and response along with the paper, I would think it satisfactory now to suggest that many of the concerns and alternative interpretations raised by myself and the other referees remain valid. Readers can read these original concerns and the rebuttal and decide for themselves how much weight to give them.

We thank the reviewer for taking the time to evaluate our revisions, and we will opt in to the transparent peer review system so that any interested reader can see the discussion of the additional alternative interpretations that we did not describe in the paper.

Reviewer #2:

Overall, the authors have addressed my concerns and have improved the manuscript. I think their model is not completely convincing, but it contributes to the currently lively discussion about the mechanism of bacterial cell division.

We thank the reviewer for taking the time to evaluate our revisions.

Reviewer #3:

In this revised manuscript, Krupka et al. address all of the questions and concerns of the reviewers and provide additional experiments that make important contributions to their working model of the role of FtsA and FtsZ in E. coli divisome assembly. The manuscript is greatly improved and will certainly attract attention to a wide readership.

In particular, the authors presented beautiful tomography data of FtsA and FtsZ organization. The resulting tomograms clearly show minirings of FtsA bound to lipid monolayers and unbundled FtsZ protofilaments bound to the membrane-distal region of the FtsA ring. However the 2D images of FtsA* + FtsZ were less convincing to myself and the other reviewers. The authors have now provided new tomographic data in a new supplemental figure (S9) that I feel clearly shows the short FtsA* oligomers proximal to the substrate and bundled FtsZ above. I completely agree that more detailed structural information using image averaging would not be possible with such heterogeneous structures. The organizational differences between the data in figure 5 (FtsZ protofilaments on top of FtsA

minirings) and the new S9 (FtsA* + FtsZ) is remarkable and strengthens their working model in the new Figure 7.

The authors also addressed and corrected all of the minor points and errors in my review. The addition of higher magnification insets (Figs 4 and 6) and arrows help to identify the structures of interest more clearly, especially to those unfamiliar to negative stain EM.

The text has also been expanded to address the issues raised about using negative stain samples for this type of high resolution work. There are limitations to the use of negative stain that the authors have now added a statement proposing the importance of future studies using cryoEM to overcome these limitations.

Although organization of FtsA minirings presented in the paper was remarkable, there is no direct evidence that minirings are actually present in cells. The authors have expanded the discussion to include a statement to future work using cryoET methods to provide important avenues for identifying these structures in a near-native state.

Overall, I feel the authors addressed all of my questions and concerns and provide a thorough and comprehensive revision that will make important contributions to the bacterial divisome field but will also be of interest to a wide readership.

We thank the reviewer for taking the time to evaluate our revisions and for the positive comments on our work.